# Truly Optimal Inverse Propensity Scoring for Off-Policy Evaluation with Multiple Loggers

## Abstract

We study off-policy evaluation (OPE) in contextual bandits with data collected from multiple logging policies. As highlighted by Agarwal et al. (2017), there seemingly exists no IPS estimator that consistently outperforms the others in this setting. We resolve this dilemma by deriving an optimal IPS estimator with sample-dependent weights that minimize variance. Through a calculus-of-variations approach, we obtain closed-form optimal weights under the unbiasedness condition, yielding an estimator that is unbiased and achieves asymptotically optimal variance. Experiments on four benchmark datasets confirm this resolution in practice, showing that our estimator consistently outperforms state-of-the-art methods with substantial relative RMSE reductions across diverse logger mixtures and numbers of logging policies.

## 1 Introduction

Off-policy evaluation (OPE) in contextual bandits is an essential causal inference tool with applications in areas such as recommender systems, medical treatment planning, and ad placement. The problem becomes substantially more challenging when data is collected under multiple logging policies—a setting that often arises in practice, where data originates from heterogeneous sources governed by different behavior policies. The presence of multiple logging policies can greatly increase the variance of OPE estimates, heightening the risk of suboptimal or unreliable policy evaluation.

Traditional OPE methods often rely on inverse propensity scoring (IPS) estimators Rosenbaum & Rubin (1983), which reweight observed data to account for discrepancies between the logging and evaluation policies. In the unstratified setting, where all data are collected under a single logging policy, IPS estimators adjust each sample using the ratio of the evaluation policy's probability to that of the logging policy for the observed action Precup et al. (2000); Bottou et al. (2013). In contrast, the stratified setting, where data are collected under multiple logging policies with fixed sample sizes, introduces substantial complexity in determining how to weight and combine data across sources.

Agarwal et al. (2017) were among the first to study OPE with multiple logging policies. They analyzed three IPS estimators based on multiple importance sampling (MIS) Cornuet et al. (2012); Elvira et al. (2019); Veach & Guibas (1995): (i) a naive estimator that averages IPS estimates computed separately for each dataset; (ii) an estimator that applies IPS to the averaged logging policy, weighted by dataset size; and (iii) an estimator that constructs a weighted average using a divergence-based weighting scheme between the logging and evaluation policies. Their analysis highlighted a key challenge: while the naive IPS estimator consistently performs the worst, the variance of the other two is highly instance-dependent, with no estimator consistently dominating. This variance sensitivity poses a practical obstacle for practitioners seeking reliable and robust OPE methods.

Building on these insights, Kallus et al. (2021) addressed this ambiguity by deriving the theoretical efficiency bound for the stratified setting—that is, the minimum achievable asymptotic variance for a broad class of OPE estimators—and proposed estimators that attain this optimal efficiency. Moreover, they showed that the efficiency bound is identical in both stratified and unstratified settings, implying that incorporating logger identities (i.e., information about which data point comes from which logging policy) is unnecessary to achieve the minimum possible variance.

In this work, we propose a novel framework for deriving optimal weight functions for the IPS estimator in the stratified OPE setting. Using the calculus of variations, we directly obtain weights that minimize the estimator's variance subject to the unbiasedness constraint. We further introduce a feasible version

of this optimally weighted IPS estimator—referred to as the optimal IPS estimator—by employing a cross-fitting approach to ensure that the optimal weights are estimated independently of the data used to compute the final estimate. Finally, we provide theoretical guarantees establishing its unbiasedness and asymptotic efficiency.

Our approach directly addresses the dilemma identified by Agarwal et al. (2017), namely that no single estimator consistently outperforms the others in the multi-logger setting, but does so differently from Kallus et al. (2021), who analyzed a broader class that includes doubly robust estimators. In contrast, we establish optimality strictly within the class of IPS estimators considered by Agarwal et al. (2017).

**Contributions**   Our main contribution is the derivation of an optimal IPS estimator for OPE in the presence of multiple logging policies. Specifically, it includes:

1. We formulate variance minimization for the generalized weighted IPS estimator (Eq. 4) as a calculus-of-variations problem. Unlike prior IPS methods, our approach allows for sample-dependent weights (Eq. 7) to achieve further variance reduction.

2. We characterize these optimal weights in closed form (Theorem 5.1). Since these quantities depend on the underlying data distribution, we propose estimating them from data. Algorithm 1 presents a practical implementation of the optimal IPS estimator using cross-fitting (van der Laan et al., 2011; Chernozhukov et al., 2018), in the spirit of Kallus et al. (2021).

3. We prove that the resulting estimator is unbiased and asymptotically optimal (Theorem 5.2). Finally, in Section 6, we empirically demonstrate that our method substantially reduces variance and consistently outperforms existing IPS baselines across four contextual bandit benchmarks. We provide code in the supplementary material to enable exact reproduction of results.

## 2   RELATED WORKS

OPE assesses the performance of counterfactual policies using logged data (Precup et al., 2000; Bottou et al., 2013).While the standard OPE setup typically assumes data collected under a single logging policy, real-world applications often involve data gathered from multiple logging policies, such as in parallel A/B testing (Agarwal et al., 2017; He et al., 2019; Kallus et al., 2021; Chen et al., 2020). This multi-logger setting, often modeled as stratified sampling with logger identities serving as strata (Kallus et al., 2021; Wooldridge, 2001), introduces unique challenges, particularly the risk of inefficiencies arising from poor-quality logging policies among the mix.

**OPE with Multiple Loggers.**   Agarwal et al. (2017) initiated the study of OPE with multiple logging policies, proposing several IPS estimators inspired by multiple importance sampling (Veach & Guibas, 1995; Cornuet et al., 2012; Elvira et al., 2019). They identified a key challenge: no single estimator consistently outperforms the others across problem instances. Kallus et al. (2021) formalized this setting by deriving a efficiency bound and showing it is achievable without incorporating logger identities. They also introduced a doubly robust estimator that asymptotically attains this bound. Related work includes off-policy learning extensions (He et al., 2019), infinite-horizon analysis (Chen et al., 2020), and generalization under selection bias (Hatt et al., 2022). Complementary to our setting, Liu et al. (2025) studied efficient multi-policy evaluation in reinforcement learning by designing a tailored behavior policy across multiple target policies, whereas we address off-policy evaluation from fixed logged bandit data with multiple loggers.

**Multiple Importance Sampling.**   Multiple importance sampling addresses the challenge of combining data from multiple sources, originally developed to reduce variance in Monte Carlo rendering (Veach & Guibas, 1995; Owen & Zhou, 2000; Elvira et al., 2019; Kondapaneni et al., 2019). Veach & Guibas (1995) introduced the balance and power heuristics, which provide robust performance under non-negative weights. Subsequent work explored more general weighting schemes (Elvira et al., 2019). In particular, Kondapaneni et al. (2019) derived provably optimal MIS weights via direct variance minimization, allowing negative weights and achieving further variance reduction, especially in defensive sampling scenarios. Our technique adapts this idea from the domain of computer graphics to OPE in contextual bandits.

## 3 PRELIMINARIES

In this section, we formulate the problem and provide background on existing methods, which will later be compared with our optimal IPS method in the experimental section.

**Notations.** For $n \in \mathbb{N}$, let $[n] := \{1, \ldots, n\}$. Indexed objects such as vectors and matrices are written in boldface: $\mathbf{v} \in \mathbb{R}^n$, $\mathbf{M} \in \mathbb{R}^{n \times n}$, where $v_i$ denotes the $i$th component of $\mathbf{v}$, and $M_{ij}$ denotes the $(i,j)$th entry of $\mathbf{M}$, that is, the element in the $i$th row and $j$th column. The notation $\mathbb{1}[\,\cdot\,]$ denotes the indicator function. For real-valued integrable functions $f, g : \Omega \to \mathbb{R}$ and a probability distribution $\mu$ over $\Omega$, we write

$$\langle f, g \rangle_\mu := \mathbb{E}_{\omega \sim \mu}[f(\omega)g(\omega)] = \mathbb{E}_\mu[fg],$$

which represents the inner product under $\mu$. Finally, $\operatorname{supp}(\mu)$ denotes the support of distribution $\mu$.

### 3.1 PROBLEM SETTING

We consider the OPE problem in the contextual bandit setting as studied in previous works Agarwal et al. (2017); He et al. (2019); Kallus et al. (2021).

A policy $\pi$ is a function mapping a context (i.e., a state) $s \in \mathcal{S}$ to a distribution over possible actions $\mathcal{A}$, formally represented as a conditional distribution $\pi(a|s)$ for each $s \in \mathcal{S}$.

An environment consists of *unknown* distributions: a context distribution $p_s(s)$ and reward distributions $p_r(r|s,a)$ over $\mathcal{R} \subseteq \mathbb{R}$ for each context-action pair $(s,a)$. The interaction between a policy and the environment begins with the environment sampling a context from $p_s(s)$, followed by the policy selecting an action $a \sim \pi(a|s)$, and concludes with a reward sampled from $p_r(r|s,a)$. This process induces a joint distribution $\mu_\pi$ over $\mathcal{S} \times \mathcal{A} \times \mathcal{R}$ with density (or mass) $p_s(s)\pi(a|s)p_r(r|s,a)$. The expected reward of a policy, denoted by $J(\pi)$, is defined as

$$J(\pi) := \mathbb{E}_{\mu_\pi}[r],$$

where $\mathbb{E}_{\mu_\pi}$ denotes the expectation with respect to $\mu_\pi$.

The goal of OPE is to estimate the expected reward $J(\pi_e)$ of a target evaluation policy $\pi_e$ using i.i.d. data $\mathcal{D} := \{(s_j, a_j, r_j)\}_{j=1}^n$ sampled from $\mu_\pi$, the distribution induced by a logging policy $\pi$, which is generally different from $\pi_e$. In this work, we consider stratified OPE with multiple logging policies $(\pi_i)_{i \in [K]}$, where the dataset is given by $\mathcal{D} := (\mathcal{D}_i)_{i \in [K]}$, and each data $\mathcal{D}_i$ consists of $n_i$ i.i.d. samples from $\mu_{\pi_i}$, that is, $\mathcal{D}_i := \{(s_{ij}, a_{ij}, r_{ij})\}_{j=1}^{n_i}$.

### 3.2 EXISTING APPROACHES

**Naive IPS.** Naive IPS addresses OPE by reweighting observed rewards from logged data to account for the probability of actions under the evaluation policy relative to the logging policy. The naive IPS estimator is defined as:

$$\hat{J}_{\text{IPS}}(\pi_e; \mathcal{D}) = \frac{1}{N} \sum_{i=1}^K \sum_{j=1}^{n_i} \frac{\pi_e(a_{ij}|s_{ij})}{\pi_i(a_{ij}|s_{ij})} r_{ij}, \tag{1}$$

where $N := \sum_{i=1}^K n_i$ is the total number of samples.

Naive IPS provides an unbiased estimate of the policy value when the propensity scores are correctly specified, making it a fundamental and widely used technique in the toolkit of OPE methods.

**Weighted IPS.** Agarwal et al. (2017) proposed an estimator based on MIS Veach & Guibas (1995), which combines probabilities from multiple logging policies to create a more efficient estimator. They introduced an IPS estimator with a balanced heuristic of MIS, defined as:

$$\hat{J}_{\text{bIPS}}(\pi_e; \mathcal{D}) = \frac{1}{N} \sum_{i=1}^K \sum_{j=1}^{n_i} \frac{\pi_e(a_{ij}|s_{ij})}{\pi_{\text{avg}}(a_{ij}|s_{ij})} r_{ij}, \tag{2}$$

where $\pi_{\text{avg}}(a|s) = \frac{1}{N} \sum_{i=1}^K n_i \pi_i(a|s)$.

Motivated by adaptive MIS Elvira et al. (2019), Agarwal et al. (2017) also introduced a weighted version of the IPS estimator, defined as:

$$\hat{J}_{\text{wIPS}}(\pi_e; \mathcal{D}) = \sum_{i=1}^{K} \lambda_i^{\star} \sum_{j=1}^{n_i} \frac{\pi_e(a_{ij}|s_{ij})}{\pi_i(a_{ij}|s_{ij})} r_{ij}, \tag{3}$$

where $\lambda_i^{\star} := \frac{1}{\sigma_r^2(\pi_e||\pi_i)} \left( \sum_{j=1}^{K} \frac{n_j}{\sigma_r^2(\pi_e||\pi_j)} \right)^{-1}$ and the divergence $\sigma_r^2$ between the target policy $\pi_e$ and a logging policy $\pi$ is given by $\sigma_r^2(\pi_e||\pi) := \text{Var}_{\mu_\pi}\left[ \frac{\pi_e(a|s)}{\pi(a|s)} r \right]$.

Unlike $\hat{J}_{\text{IPS}}$ and $\hat{J}_{\text{bIPS}}$, the estimator $\hat{J}_{\text{wIPS}}$ is not directly feasible in practice since the weights $\lambda_i^{\star}$ must first be estimated from data. To address this, Kallus et al. (2021) proposed a feasible version of $\hat{J}_{\text{wIPS}}$ using a cross-fitting technique van der Laan et al. (2011); Chernozhukov et al. (2018).

## 4 OPTIMAL IPS ESTIMATOR FOR OPE WITH MULTIPLE LOGGERS

In this section, we present our main contribution: the optimal IPS estimator for OPE with multiple loggers, denoted $\hat{J}_{\text{oIPS}}$ and outlined in Algorithm 1.

### 4.1 GENERALIZED WEIGHTED IPS ESTIMATOR CLASS $\Gamma_{\text{gwIPS}}$.

We show that our estimator is optimal within a general class of weighted IPS estimators. Let $\mathbf{w} := (w_i)_{i \in [K]}$, where each $w_i : \mathcal{S} \times \mathcal{A} \times \mathcal{R} \to \mathbb{R}$ is a sample-dependent weight. The generalized weighted IPS estimator class, $\Gamma_{\text{gwIPS}}$, is defined as the set of all estimators of the following form:

$$\hat{J}(\pi_e, \mathbf{w}; \mathcal{D}) := \sum_{i=1}^{K} \frac{1}{n_i} \sum_{j=1}^{n_i} \frac{w_i(s_{ij}, a_{ij}, r_{ij}) \pi_e(a_{ij}|s_{ij}) r_{ij}}{\pi_i(a_{ij}|s_{ij})}. \tag{4}$$

An estimator in $\Gamma_{\text{gwIPS}}$ is unbiased if the weights $w_i$ satisfy the following conditions. The first is the *normalization condition*:

$$\sum_{i=1}^{K} w_i(s, a, r) = 1, \quad \forall (s, a, r) \in \text{supp}(\mu_{\pi_e}), \tag{5}$$

and the second is the *dominance condition*:

$$\forall i \in [K], \quad \pi_i(a|s) = 0 \implies w_i(s, a, r) = 0. \tag{6}$$

Under these constraints, a straightforward calculation shows that $\hat{J}(\pi_e, \mathbf{w}; \mathcal{D})$ is unbiased, as stated in the following theorem (full proof in Appendix A).

**Theorem 4.1.** *For any weights* $\mathbf{w} : \Omega \to \mathbb{R}$ *that satisfies conditions* (5) *and* (6),

$$\mathbb{E}[\hat{J}(\pi_e, \mathbf{w}; \mathcal{D})] = J(\pi_e),$$

*where the first expectation is over the randomness of the sampling of the dataset* $\mathcal{D} := (\mathcal{D}_i)_{i \in [K]}$.

Subject to conditions (5) and (6), we seek weights $\mathbf{w}$ that minimize the variance $\text{Var}[\hat{J}(\pi_e, \mathbf{w}; \mathcal{D})]$. This variance can be viewed as a functional over vector-valued functions $\mathbf{w} : \mathcal{S} \times \mathcal{A} \times \mathcal{R} \to \mathbb{R}^K$, which allows us to derive the optimal weights $\mathbf{w}^{\star}(s, a, r)$ using the calculus of variations. Theorem 5.1 shows that these optimal weights $\mathbf{w}^{\star}$ minimizing $\text{Var}[\hat{J}(\pi_e, \mathbf{w}; \mathcal{D})]$ take the form of

$$w_i^{\star}(s, a, r) = \frac{\alpha_i \pi_i(a|s)}{r \pi_e(a|s)} + \frac{n_i \pi_i(a|s)}{\sum_{k=1}^{K} n_k \pi_k(a|s)} \left( 1 - \frac{\sum_{k=1}^{K} \alpha_k \pi_k(a|s)}{r \pi_e(a|s)} \right), \tag{7}$$

the internal weight vector $\boldsymbol{\alpha} := (\alpha_i)_{i \in [K]}$ is a solution to the linear system $\mathbf{T}\boldsymbol{\alpha} = \mathbf{c}$, with the entries of $\mathbf{T} \in \mathbb{R}^{K \times K}$ and $\mathbf{c} \in \mathbb{R}^K$ given by

$$T_{ij} := \mathbb{E}_{\mu_{\pi_i}}\left[ \frac{n_i \pi_j(a|s)}{\sum_{k=1}^{K} n_k \pi_k(a|s)} \right], \quad c_i := \mathbb{E}_{\mu_{\pi_i}}\left[ \frac{r \, n_i \, \pi_e(a|s)}{\sum_{k=1}^{K} n_k \pi_k(a|s)} \right]. \tag{8}$$

We refer to $\mathbf{T}$ as the technique matrix and $\mathbf{c}$ as the contribution vector.

---

**Algorithm 1** Optimal IPS for Off-Policy Evaluation

---

1: **Input:** Dataset $\mathcal{D} := (\mathcal{D}_i)_{i \in [K]}$ where

$$\mathcal{D}_i := \{(s_{ij}, a_{ij}, r_{ij})\}_{j=1}^{n_i},$$

being i.i.d. samples from $\mu_{\pi_i}$.

2: **Output:** Estimated policy value $\hat{J}_{\text{oIPS}}(\pi_e; \mathcal{D})$.

3: For each $i \in [K]$, let $\{\mathcal{D}_i^{(z)}\}_{z \in [Z]}$ be a $Z$-fold random partition of $\mathcal{D}_i$, such that the sizes of the folds, $|\mathcal{D}_i^{(z)}|$, are approximately equal.

4: For each $z \in [Z]$, let $\mathcal{D}^{(z)} := \{\mathcal{D}_i^{(z)}\}_{i \in [K]}$ and $\mathcal{D}^{(-z)} := \{\mathcal{D}_i \setminus \mathcal{D}_i^{(z)}\}_{i \in [K]}$.

5: **for** $z \in [Z]$ **do**

6:    Estimate the technique matrix $\hat{\mathbf{T}}^{(z)}$ and the contribution vector $\hat{\mathbf{c}}^{(z)}$ using $\mathcal{D}^{(-z)}$ via (9).

7:    Let $\hat{\boldsymbol{\alpha}}^{(z)} := (\hat{\mathbf{T}}^{(z)\top} \hat{\mathbf{T}}^{(z)})^{-1} \hat{\mathbf{T}}^{(z)\top} \hat{\mathbf{c}}^{(z)}$ be the OLS estimator.

8:    Compute $\hat{J}^{(z)}$ using $\hat{\boldsymbol{\alpha}}^{(z)}$ and $\mathcal{D}^{(z)}$ via (11).

9: **end for**

10: **Return** $\hat{J}_{\text{oIPS}}(\pi_e; \mathcal{D}) := \frac{1}{N} \sum_{z \in [Z]} |\mathcal{D}^{(z)}| \hat{J}^{(z)}$, where $N := \sum_{i=1}^{K} n_i$.

---

Unfortunately, $\mathbf{T}$ and $\mathbf{c}$ depend on the unknown contextual bandit environment (i.e., $p_s$ and $p_r$), so $\mathbf{w}^\star$ cannot be computed directly. This is analogous to the weighted IPS estimator (3), where the optimal coefficients $\lambda_i^\star$ also depend on the environment, rendering the naive form infeasible. Instead, these quantities must be estimated from data via cross-fitting, which yields a feasible estimator Kallus et al. (2021). Following this approach, in the next section we present a feasible version of the ideal optimal IPS estimator $\hat{J}(\pi_e, \mathbf{w}^\star; \mathcal{D})$, denoted $\hat{J}_{\text{oIPS}}$.

### 4.2 Asymptotically Optimal IPS Estimator $\hat{J}_{\text{oIPS}}$

In this section, we outline how cross-fitting can be used to derive a feasible optimal estimator $\hat{J}_{\text{oIPS}}$. Specifically, we employ the cross-fitting strategy van der Laan et al. (2011); Chernozhukov et al. (2018): one part of the data is used to estimate $\mathbf{T}$ and $\mathbf{c}$ (and hence the optimal weights), while the other is used to compute the weighted IPS estimator with these estimated weights. The complete procedure for computing $\hat{J}_{\text{oIPS}}$ is provided in Algorithm 1.

**Computing $\hat{J}_{\text{oIPS}}$.** For each $i \in [K]$, let $\{\mathcal{D}_i^{(z)}\}_{z \in [Z]}$ be a $Z$-fold random partition of $\mathcal{D}_i$, such that the sizes of the folds $|\mathcal{D}_i^{(z)}|$ are approximately equal. Then, for each $z \in [Z]$, we estimate $\mathbf{T}$ and $\mathbf{c}$ using $\hat{\mathbf{T}}^{(z)}$ and $\hat{\mathbf{c}}^{(z)}$, respectively—obtained as sample averages of (8) over the rest of *zth* fold data, $\mathcal{D}^{(-z)} := \{\mathcal{D}_i \setminus \mathcal{D}_i^{(z)}\}_{i \in [K]}$.

For each $i \in [K]$ and $z \in [Z]$, let $n_i^{(z)}$ denote the size of $\mathcal{D}_i^{(z)}$. Assign indices $j = 1, \ldots, n_i^{(z)}$ to the data points in $\mathcal{D}_i^{(z)}$, and indices $j = n_i^{(z)} + 1, \ldots, n_i$ to the remaining data points in $\mathcal{D}_i \setminus \mathcal{D}_i^{(z)}$. Then, the estimators $\hat{\mathbf{T}}^{(z)} \in \mathbb{R}^{K \times K}$, $\hat{\mathbf{c}}^{(z)} \in \mathbb{R}^K$ of $\mathbf{T}$ and $\mathbf{c}$, respectively, are given by

$$\hat{T}_{ij}^{(z)} := \frac{1}{n_i - n_i^{(z)}} \sum_{l=n_i^{(z)}+1}^{n_i} \frac{n_i \, \pi_j(a_{il}|s_{il})}{\sum_{k=1}^K n_k \, \pi_k(a_{il}|s_{il})},$$

$$\hat{c}_i^{(z)} := \frac{1}{n_i - n_i^{(z)}} \sum_{l=n_i^{(z)}+1}^{n_i} \frac{r_{il} \, n_i \, \pi_e(a_{il}|s_{il})}{\sum_{k=1}^K n_k \, \pi_k(a_{il}|s_{il})}. \tag{9}$$

We use the above estimates as drop-in replacements for $\mathbf{T}$ and $\mathbf{c}$ when computing $\boldsymbol{\alpha}$, and hence $w^\star$. Specifically, let $N := \sum_{i=1}^{K} n_i$ and $N^{(z)} := \sum_{i=1}^{K} n^{(z)}i$. The optimal IPS estimator is defined as

$$\hat{J}_{\text{oIPS}}(\pi_e; \mathcal{D}) := \frac{1}{N} \sum_{z \in [Z]} N^{(z)} \hat{J}^{(z)}, \quad \hat{J}^{(z)} := \sum_{i=1}^{K} \frac{1}{n_i^{(z)}} \sum_{j=1}^{n_i^{(z)}} \hat{w}_{ij}^{(z)} r_{ij} \frac{\pi_e(a_{ij}|s_{ij})}{\pi_i(a_{ij}|s_{ij})}, \quad (10)$$

where the weight $\hat{w}_{ij}^{(z)} := \hat{w}_i^{(z)}(s_{ij}, a_{ij}, r_{ij})$ is given by

$$\hat{w}_i^{(z)}(s, a, r) := \frac{\hat{\alpha}_i^{(z)} \pi_i(a|s)}{r \, \pi_e(a|s)} + \frac{n_i^{(z)} \pi_i(a|s)}{\sum_{k=1}^{K} n_k^{(z)} \pi_k(a|s)} \left( 1 - \frac{\sum_{k=1}^{K} \hat{\alpha}_k^{(z)} \pi_k(a|s)}{r \, \pi_e(a|s)} \right),$$

where $\hat{\boldsymbol{\alpha}}^{(z)} = (\hat{\alpha}_i^{(z)})_{i \in [K]}$ is the OLS estimator, defined as $\hat{\boldsymbol{\alpha}}^{(z)} := (\hat{\mathbf{T}}^{(z)\top} \hat{\mathbf{T}}^{(z)})^{-1} \hat{\mathbf{T}}^{(z)\top} \hat{\mathbf{c}}^{(z)}$.

By plugging the above directly into (10), $\hat{J}^{(z)}$ simplifies to

$$\hat{J}^{(z)} = \sum_{i=1}^{K} \hat{\alpha}_i^{(z)} + \sum_{i=1}^{K} \sum_{j=1}^{n_i^{(z)}} \frac{r_{ij} \pi_e(a_{ij}|s_{ij})}{\sum_{k=1}^{K} n_k^{(z)} \pi_k(a_{ij}|s_{ij})} - \frac{\sum_{k=1}^{K} \hat{\alpha}_k^{(z)} \pi_k(a_{ij}|s_{ij})}{\sum_{k=1}^{K} n_k^{(z)} \pi_k(a_{ij}|s_{ij})}. \quad (11)$$

Notably, the second term recovers the balanced IPS estimator over $\mathcal{D}^{(z)}$.

For the optimal IPS estimator with cross-fitting, we prove in Theorem 5.2 that $\hat{J}_{\text{oIPS}}(\pi_e; \mathcal{D})$, as defined above, remains unbiased, consistent, and asymptotically efficient.

**When Logging Policies are Unknown.** When logging policies are unknown, we substitute consistent estimators of the propensities $\pi_i(a_{jk}|s_{ik})$ when computing $\hat{J}_{\text{oIPS}}$. The estimation procedure for unknown logging policies is described in Section 6. We show that our estimator achieves good and stable practical performance even when using estimated logging propensities.

## 5 ANALYSIS

In this section, we provide a theoretical analysis of our optimal IPS estimator $\hat{J}_{\text{oIPS}}$. Specifically, we prove the optimality of the generalized weights $\mathbf{w}^\star$ given by (7) and establish that $\hat{J}_{\text{oIPS}}(\pi_e; \mathcal{D})$ is consistent and asymptotically normal with the optimal variance $\text{Var}[\hat{J}(\pi_e, \mathbf{w}^\star; \mathcal{D})]$.

### 5.1 TECHNICAL ASSUMPTION

Before proceeding, we state a technical assumption on the logging and evaluation policies.
**Assumption 5.1.** *For almost all $s \in \mathcal{S}$, $\cup_{i \in [K]} \text{supp}(\pi_i(a|s)) \subseteq \text{supp}(\pi_e(a|s))$.*

This assumption is used only to streamline the proofs and is not required for the final statements. It can be removed via a standard continuity (smoothing) argument: for any $\pi_e$, define the $\varepsilon$–perturbed policy $\pi_e^\varepsilon := (1 - \varepsilon), \pi_e + \varepsilon, \pi_u$, where $\pi_u$ is uniform over actions. Then $\pi_e^\varepsilon$ has full support and satisfies Assumption 5.1. Results proven under Assumption 5.1 therefore hold for $\pi_e^\varepsilon$; letting $\varepsilon \to 0$ and using continuity of our estimator and objective in $\pi_e$, the optimality conclusions extend to the original $\pi_e$.

Notably, our estimator *does not rely on* the weak or strong overlap conditions of Kallus et al. (2021), such as $\pi_e(a|s) \subseteq \cup_{i \in [K]} \pi_i(a|s)$ (weak) or $\pi_e(a|s) \subseteq \cap_{i \in [K]} \pi_i(a|s)$ (strong).

### 5.2 CHARACTERIZING VARIANCE OF ESTIMATORS IN $\Gamma_{\text{gwIPS}}$

We begin our analysis with the following theorem, which characterizes the variance of $\hat{J}(\pi_e, \mathbf{w}; \mathcal{D})$ and enables computation of the functional derivative with respect to $\mathbf{w}$. The full proof is provided in Appendix A.

**Lemma 5.1.** *Let $\Omega := \mathcal{S} \times \mathcal{A} \times \mathcal{R}$. For any weights $\mathbf{w} : \Omega \to \mathbb{R}$ that satisfies conditions (5) and (6),*

$$\text{Var}\big[\hat{J}(\pi_e, \mathbf{w}; \mathcal{D})\big] = \sum_{i=1}^{K} \frac{1}{n_i} \left( \left\langle w_i^2, \frac{\pi_e \Pi_r^2}{\pi_i} \right\rangle_{\mu_{\pi_e}} - \langle w_i, \Pi_r \rangle_{\mu_{\pi_e}}^2 \right),$$

*where $\Pi_r(\omega) := r$ for all $\omega := (s, a, r) \in \Omega$.*

### 5.3 DERIVATION OF OPTIMAL GENERALIZED WEIGHTS.

We aim to find optimal weights $\mathbf{w}^\star$, defined as the solution to the following variance minimization problem, subject to constraints that guarantee unbiasedness:

$$\underset{\mathbf{w}}{\text{minimize}} \quad \text{Var}\big[\hat{J}(\pi_e, \mathbf{w}; \mathcal{D})\big] \tag{12}$$

$$\text{subject to} \quad \sum_{i=1}^{K} w_i(\omega) = 1, \quad \forall \omega \in \text{supp}\,(\mu_{\pi_e}), \text{ and}$$

$$\pi_i(a|s) = 0 \implies w_i(s, a, r) = 0, \quad \forall i \in [K].$$

With this formulation, we provide a characterization of the optimal weights $\mathbf{w}^\star : \Omega := \mathcal{S} \times \mathcal{A} \times \mathcal{R} \to \mathbb{R}$ in the following theorem whose proof is provided in Appendix A.

**Theorem 5.1.** *For logging and evaluation policies $(\pi_i)_{i \in [K]}$ and $\pi_e$ that satisfy Assumption 5.1, let $\mathbf{T} \in \mathbb{R}^{K \times K}$ be the technique matrix and $\mathbf{c} \in \mathbb{R}^K$ be the contribution vector defined in (8). Let $\boldsymbol{\alpha}$ be any solution to $\mathbf{T}\boldsymbol{\alpha} = \mathbf{c}$, if one exists. Then, with respect to such $\boldsymbol{\alpha}$, the weight vector $\mathbf{w}^\star$ given by (7) solves the optimization problem (12).*

**Unbiasedness and Asymptotic Efficiency of $\hat{J}_{\text{oIPS}}$.** We show that $\hat{J}_{\text{oIPS}}$, the feasible version of $\hat{J}(\pi_e, \mathbf{w}^\star; \mathcal{D})$ given in Algorithm 1, remains unbiased. Furthermore, the estimator is consistent and asymptotically achieves the optimal variance $\text{Var}[\hat{J}(\pi_e, \mathbf{w}^\star; \mathcal{D})]$. By the asymptotic regime in the multi-logger setting, we follow the scaling of Kallus et al. (2021): for each logger $i \in [K]$, let $n_i' = mn_i$, and consider the limit $m \to \infty$.

**Theorem 5.2.** *Let $N' := \sum_{i=1}^{K} n_i'$ with $n_i' = mn_i'$. Then, $\hat{J}_{\text{oIPS}}$ is unbiased and*

$$\sqrt{N'} \left( \hat{J}_{\text{oIPS}}(\pi_e; \mathcal{D}) - J(\pi_e) \right) \xrightarrow{d} \mathcal{N} \left( 0, N \, \text{Var}[\hat{J}(\pi_e, \mathbf{w}^\star; \mathcal{D})] \right)$$

*as $m \to \infty$, where $N := \sum_{i=1}^{K} n_i$.*

In other words, $\hat{J}_{\text{oIPS}}$ is asymptotically normal with the optimal variance $\text{Var}[\hat{J}(\pi_e, \mathbf{w}^\star; \mathcal{D})]$, while higher-order variances are asymptotically negligible. The proof is provided in Appendix A.

## 6 NUMERICAL EXPERIMENTS

In this section, we conduct numerical benchmarks to evaluate the performance of our optimal IPS estimator and compare it against existing methods.

### 6.1 BENCHMARK DATASETS AND EXPERIMENT SETUP

To evaluate the efficacy of different OPE methods under multiple logging policies, we adopt experimental setups from prior work on multi-logger OPE (Agarwal et al., 2017; Kallus et al., 2021). We use four datasets

Table 1: Dataset Statistics

| Dataset Name | OptDigits | SatImage | PenDigits | Letter |
|---|---|---|---|---|
| #Classes ($l$) | 10 | 6 | 10 | 26 |
| #Data ($n$) | 5620 | 6435 | 10992 | 20000 |

from the UCI Machine Learning Repository (Table 1): *OptDigits* Alpaydin & Kaynak (1998), *SatImage* Srinivasan (1993), *PenDigits* Alpaydin & Alimoglu (1998), and *Letter* Slate (1991) (Table 1). Each classification dataset is converted into a contextual bandit dataset by treating labels as actions and assigning rewards of 1 for correct classifier predictions and 0 otherwise, as described in more detail below.

**Mapping Classification to Contextual Bandits.** We map the multi-class classification problem to a contextual bandit setting. In multi-class classification with deterministic labels, data is distributed as $(s, y(s))$, where $s \sim p_s(s)$ is the marginal data distribution. We treat $p_s$ as the state distribution in the contextual bandit framework, and define the deterministic reward as $r(s, a) := \mathbb{1}[y(s) = a]$.

To generate bandit feedback from a dataset $\{(s_i, y_i)\}_{i=1}^{n}$, we train a deterministic classifier $\hat{y}$ on 30% of the data and construct the corresponding policy $\pi_{\text{det}}(a|s) := \mathbb{1}[a = \hat{y}(s)]$. Specifically, we use logistic regression on 30% of the data to obtain $\pi_{\text{det}}$. The evaluation policy is set to $\pi_e = \pi_{\text{det}}$, while the logging policies are defined as mixtures: $\pi_1 = 0.95\pi_{\text{det}} + 0.05\pi_u$ and $\pi_2 = 0.05\pi_{\text{det}} + 0.95\pi_u$, where $\pi_u$ is the uniform random policy over actions. This setup creates different levels of overlap

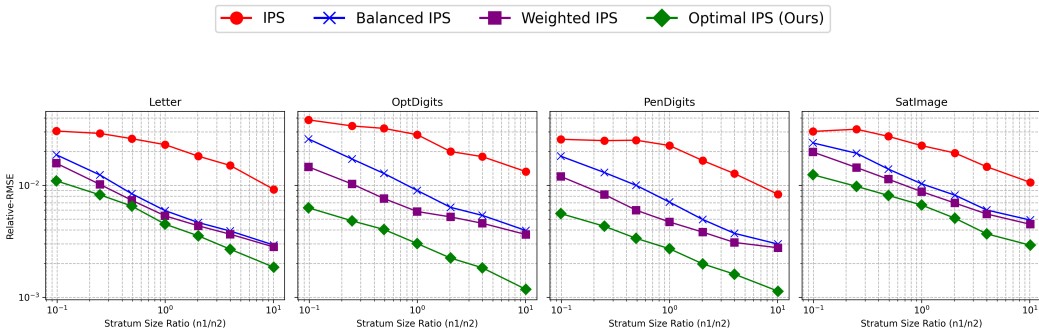

Figure 1: Relative-RMSEs (13), computed over $M = 200$ independent experiments, are reported across different datasets and estimators under the setting where the logging policies are known. Our proposed estimator $\hat{J}_{\text{oIPS}}$ is implemented with $Z = 5$-fold cross-fitting. The $x$-axis denotes the stratum size ratio $(n_1/n_2)$ on a logarithmic scale, while the $y$-axis shows the relative RMSE, also on a logarithmic scale. Lower values indicate better performance.

between the logging and evaluation policies, enabling robustness evaluation of estimators under varying conditions.

The classification accuracy of $\hat{y}$ on the remaining 70% of the data (the evaluation set) is taken as the ground-truth value of the policy $\pi_e = \pi_{\text{det}}$. We partition the evaluation set into $\mathcal{D}_1$ (generated by $\pi_1$) and $\mathcal{D}_2$ (generated by $\pi_2$) according to the ratio $n_1/n_2 = \rho_1/(1 - \rho_1)$, with $\rho_1/(1 - \rho_1) \in \{0.1, 0.25, 0.5, 1, 2, 4, 10\}$. This corresponds to $100 \times \rho_1/(1 - \rho_1)$ percent of the evaluation set being assigned to $\mathcal{D}_1$, while smaller ratios yield larger $\mathcal{D}_2$, generated by a logging policy less similar to the evaluation policy. Using this partitioned evaluation set, we benchmark estimators $\hat{J}$ in estimating the true policy value $J$.

We evaluate estimators with the Relative Root Mean Squared Error (Relative-RMSE), which quantifies the accuracy of an estimator in predicting the expected reward of the target policy. Suppose we run $M$ experiments with independently sampled datasets. The Relative-RMSE of an estimator $\hat{J}$ is

$$\text{Relative-RMSE}(\hat{J}) := \frac{1}{J(\pi_e)} \sqrt{\frac{1}{M} \sum_{m=1}^{M} \left( J(\pi_e) - \hat{J}_m \right)^2}, \tag{13}$$

where $\hat{J}_m$ denotes the estimated policy value in the $m$-th experiment.

**When Logging Policies are Unknown.**  When logging policies are unknown, we estimate each logger's behavior policy $\hat{\pi}_i$ using per-logger, 2-fold cross-fitted multinomial logistic regression. Out-of-fold model provides per-sample propensities $\hat{\pi}_i(a_{ij}|s_{ij})$, while a model trained on all data from logger $i$ is used to score every sample $(s_{jk}, a_{jk})$, yielding $\hat{\pi}_i(a_{jk}|s_{jk})$. To stabilize importance weights, we blend predicted action distributions with a 1% uniform component, clip propensities below $10^{-3}$, and default to the uniform policy for degenerate strata (e.g., too few samples or only one observed class) before plugging $\hat{\pi}$ into the IPS estimators.

**Estimators Considered.**  We compare the performance of our optimal IPS estimator $\hat{J}_{\text{oIPS}}$, with following baseline estimators:

- *Naive IPS $\hat{J}_{\text{IPS}}$ (Eq. 1).* The naive IPS estimator reweights observed rewards from logged data based on the probability of the actions under the evaluation policy relative to the logging policy. This method provides an unbiased estimate of the policy value when the propensity scores are correctly specified.

- *Balanced IPS $\hat{J}_{\text{bIPS}}$ (Eq. 2).* The bIPS estimator, introduced by Agarwal et al. (2017), employs a balanced heuristic inspired by multiple importance sampling Veach & Guibas (1995), aiming to improve performance across multiple logging policies.

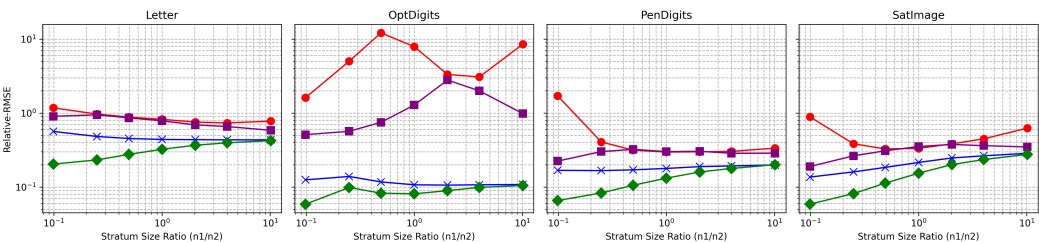

Figure 2: Relative-RMSEs (13), computed over $M = 200$ independent experiments, are reported across different datasets and estimators under the setting where the logging policies are estimated.

- *Weighted IPS $\hat{J}_{\text{wIPS}}$ (Eq. 3)*. The wIPS estimator, motivated by adaptive MIS Elvira et al. (2019) and proposed by Agarwal et al. (2017), combines samples from different logging policies using adaptive weights. For our experiments, we use the feasible version of $\hat{J}_{\text{wIPS}}$ Kallus et al. (2021), which computes empirical adaptive weights.

## 6.2 EXPERIMENTAL RESULTS.

The Relative-RMSEs on the *Letter*, *OptDigits*, *PenDigits*, and *SatImage* datasets, for varying values of $n_1/n_2$, are presented in Figures 1 and 2, with each value computed from $M = 200$ independent experiments. We use the $Z = 5$-fold cross-fitting version of our estimator $\hat{J}_{\text{oIPS}}$.

When the logging policies are known (Figure 1), the optimal IPS estimator consistently outperforms all baselines across datasets and stratum size ratios, confirming its theoretical optimality in practice. Its Relative-RMSE remains low and stable, demonstrating both accuracy and robustness. In contrast, Naive IPS and Balanced IPS exhibit higher Relative-RMSEs, particularly at smaller stratum size ratios, underscoring their sensitivity to mismatches between logging and evaluation policies.

When logging policies are unknown, we observe the same qualitative pattern as in the known-policy setting: our optimal IPS achieves the lowest Relative-RMSE across datasets and most stratum ratios, often by a wide margin. This advantage arises from $\hat{J}_{\text{oIPS}}$'s sample-dependent weights, learned via cross-fitting of the technique matrix and contribution vector, which robustly balance strata even when propensity estimates are noisy. By contrast, $\hat{J}_{\text{wIPS}}$ underperforms $\hat{J}_{\text{bIPS}}$ because its adaptive mixture weights rely on plug-in divergence estimates, which are sensitive to logging-policy estimation error. In these cases, estimation noise leads to suboptimal weighting across loggers and inflated variance, whereas the balance heuristic in $\hat{J}_{\text{bIPS}}$ remains more stable.

Overall, the optimal IPS estimator achieves superior performance to other IPS estimators, whether logging policies are known or estimated. Its lower variance and robustness across diverse scenarios underscore its effectiveness in handling various logging-policy conditions. We supplement this conclusion in Appendix B and C, where we report additional experiments with more than two loggers and varying mixture ratios. These results confirm that the asymptotic optimality guarantee of the optimal IPS estimator extends to practical regimes.

## 7 CONCLUSION

This study tackles the challenge of OPE in scenarios where data is generated by multiple logging policies. We proposed the optimal IPS method, which leverages sample-dependent weights to achieve asymptotically optimal variance. Through numerical experiments on four benchmark datasets, we showed that our estimator consistently outperforms baseline methods, both when logging policies are known and when they are estimated. These results validate the asymptotic optimality of our method and demonstrate its practical advantages.

Future work could focus on deriving finite-sample analyses and generalization bounds for the optimal IPS estimators, providing a precise understanding of their behavior under limited data. Additionally, exploring the integration of the optimal IPS framework with doubly robust methods and advanced robust estimation techniques could further improve its robustness and efficiency.

ETHICS STATEMENT

This study examines off-policy evaluation in contextual bandits using only publicly available benchmark datasets. It does not involve human-subject research, interventions, or protected health information. No personally identifiable data were collected or processed, so institutional review board (IRB) approval was not required. All datasets were used in accordance with their licenses/terms, and no re-identification attempts were made.

While our methods aim to reduce variance in policy evaluation, they could inform decisions in sensitive domains (e.g., recommendations, healthcare, advertising). To mitigate potential harms, we stress that our estimator is not a substitute for domain-specific oversight and should be paired with uncertainty reporting, distribution-shift checks, subgroup bias audits, and human review before deployment. We discourage use in settings where such safeguards cannot be met.

We release code and experimental configurations to support reproducibility and responsible verification of results, but do not release any data that could compromise privacy. The authors declare no financial or personal conflicts of interest and no external sponsorship that could unduly influence the work.

REPRODUCIBILITY STATEMENT

We take reproducibility seriously and provide all necessary resources to replicate our results. The complete algorithmic procedure—including cross-fitting, construction of the technique matrix, and contribution vector—is presented in Algorithm 1. Theoretical guarantees (unbiasedness, consistency, and asymptotic efficiency) are stated in Theorems 4.1–5.2, with full proofs deferred to Appendix A. Experimental details, including datasets, data-to-bandit conversions, train/evaluation splits, and evaluation metrics (Relative-RMSE), are described in Section 6.

Our anonymously submitted codebase implements optimal-IPS weights together with a cross-fitting pipeline. It includes scripts to reproduce all figures, configuration files for each dataset, and end-to-end instructions covering raw data preparation through evaluation. Environment files (e.g., Dockerfile, requirements.txt) are provided to standardize dependencies.

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

## A  OMITTED PROOFS

In this section, we present detailed proofs of the theorems and lemmas stated in the main text.

**Theorem 4.1.** *For any weights* $\mathbf{w} : \Omega \to \mathbb{R}$ *that satisfies conditions* (5) *and* (6),

$$\mathbb{E}[\hat{J}(\pi_e, \mathbf{w}; \mathcal{D})] = J(\pi_e),$$

*where the first expectation is over the randomness of the sampling of the dataset* $\mathcal{D} := (\mathcal{D}_i)_{i \in [K]}$.

*Proof of Theorem 4.1.* Let $\Omega := \mathcal{S} \times \mathcal{A} \times \mathcal{R}$ and $\omega := (s, a, r) \in \Omega$ ald $\mu_{\pi_i}$ be the probability distribution with density (mass) $p_r(r|s, a)\pi_i(a|s)p_s(s)$. Recall the general weighted MIS estimator,

$$\hat{J}(\pi_e, \mathbf{w}; \mathcal{D}) := \sum_{i=1}^{K} \frac{1}{n_i} \sum_{j=1}^{n_i} \frac{w_i(s_{ij}, a_{ij}, r_{ij})\pi_e(a_{ij}|s_{ij})r_{ij}}{\pi_i(a_{ij}|s_{ij})} := \sum_{i=1}^{K} \frac{1}{n_i} \sum_{j=1}^{n_i} \hat{J}_{ij}(w_i; \mathcal{D}_i).$$

Then,

$$
\begin{aligned}
\mathbb{E}[\hat{J}(\pi_e, \mathbf{w}; \mathcal{D})] &= \sum_{i=1}^{K} \frac{1}{n_i} \sum_{j=1}^{n_i} \mathbb{E}[\hat{J}_{ij}(w_i; \mathcal{D}_i)] \\
&= \sum_{i=1}^{K} \frac{1}{n_i} \sum_{j=1}^{n_i} \int_{\text{supp}(\mu_{\pi_i})} \frac{w_i(\omega)\pi_e(a|s)r}{\pi_i(a|s)} \, d\mu_{\pi_i}(\omega) \\
&= \sum_{i=1}^{K} \frac{1}{n_i} \sum_{j=1}^{n_i} \int_{\text{supp}(\mu_{\pi_i})} \frac{w_i(\omega)\pi_e(a|s)r}{\pi_i(a|s)} p_r(r|s, a)\pi_i(a|s)p_s(s) \, d\mu(\omega) \\
&= \sum_{i=1}^{K} \int_{\text{supp}(\mu_{\pi_i})} w_i(\omega) \cdot r \cdot p_r(r|s, a)\pi_e(a|s)p_s(s) \, d\mu(\omega) \\
&= \sum_{i=1}^{K} \int_{\Omega} w_i(\omega) \cdot r \cdot p_r(r|s, a)\pi_e(a|s)p_s(s) \, d\mu(\omega) \\
&= \int_{\Omega} \sum_{i=1}^{K} w_i(\omega) \cdot r \cdot p_r(r|s, a)\pi_e(a|s)p_s(s) \, d\mu(\omega) \\
&= \int_{\Omega} r \cdot p_r(r|s, a)\pi_e(a|s)p_s(s) \, d\mu(\omega) \\
&= \mathbb{E}_{\pi_e}[r].
\end{aligned}
$$

where the fifth line follows from the condition $\pi_i(a|s) = 0 \implies w_i(s, a, r) = 0$ for all $i \in [K]$. $\square$

**Lemma 5.1.** *Let* $\Omega := \mathcal{S} \times \mathcal{A} \times \mathcal{R}$. *For any weights* $\mathbf{w} : \Omega \to \mathbb{R}$ *that satisfies conditions* (5) *and* (6),

$$\text{Var}[\hat{J}(\pi_e, \mathbf{w}; \mathcal{D})] = \sum_{i=1}^{K} \frac{1}{n_i} \left( \left\langle w_i^2, \frac{\pi_e \Pi_r^2}{\pi_i} \right\rangle_{\mu_{\pi_e}} - \langle w_i, \Pi_r \rangle_{\mu_{\pi_e}}^2 \right),$$

*where* $\Pi_r(\omega) := r$ *for all* $\omega := (s, a, r) \in \Omega$.

*Proof of Lemma 5.1.* As $\hat{J}_{ij}(w_i; \mathcal{D}_i)$ are independent to each other,

$$
\begin{aligned}
\text{Var}[\hat{J}(\pi_e, \mathbf{w}; \mathcal{D})] &= \text{Var} \left[ \sum_{i=1}^{K} \frac{1}{n_i} \sum_{j=1}^{n_i} \hat{J}_{ij}(w_i; \mathcal{D}_i) \right] = \sum_{i=1}^{K} \frac{1}{n_i^2} \sum_{j=1}^{n_i} \text{Var}[\hat{J}_{ij}(w_i; \mathcal{D}_i)] \\
&= \sum_{i=1}^{K} \frac{1}{n_i^2} \sum_{j=1}^{n_i} \mathbb{E}[\hat{J}_{ij}^2(w_i; \mathcal{D}_i)] - \sum_{i=1}^{K} \frac{1}{n_i^2} \sum_{j=1}^{n_i} \mathbb{E}[\hat{J}_{ij}(w_i; \mathcal{D}_i)]^2.
\end{aligned}
$$

The first term is

$$
\sum_{i=1}^{K} \frac{1}{n_i^2} \sum_{j=1}^{n_i} \mathbb{E}[\hat{J}_{ij}^2(w_i; \mathcal{D}_i)] = \sum_{i=1}^{K} \frac{1}{n_i^2} \sum_{j=1}^{n_i} \int_{\Omega} \frac{w_i^2(\omega) \pi_e^2(a|s) r^2}{\pi_i^2(a|s)} p_r(r|s,a) \pi_i(a|s) p_s(s) d\mu(\omega)
$$

$$
= \sum_{i=1}^{K} \frac{1}{n_i} \int_{\Omega} \frac{w_i^2(\omega) \pi_e(a|s) r^2}{\pi_i(a|s)} p_r(r|s,a) \pi_e(a|s) p_s(s) d\mu(\omega)
$$

$$
= \sum_{i=1}^{K} \frac{1}{n_i} \int_{\Omega} w_i^2(\omega) \cdot \frac{\pi_e(a|s) r^2}{\pi_i(a|s)} d\mu_{\pi_e}(\omega)
$$

$$
= \sum_{i=1}^{K} \frac{1}{n_i} \left\langle w_i^2, \frac{\pi_e \Pi_r^2}{\pi_i} \right\rangle_{\mu_{\pi_e}}.
$$

The second term is

$$
\sum_{i=1}^{K} \frac{1}{n_i^2} \sum_{j=1}^{n_i} \mathbb{E}[\hat{J}_{ij}(w_i; \mathcal{D}_i)]^2 = \sum_{i=1}^{K} \frac{1}{n_i^2} \sum_{j=1}^{n_i} \left( \int_{\Omega} \frac{w_i(\omega) \pi_e(a|s) r}{\pi_i(a|s)} \cdot p_r(r|s,a) \pi_i(a|s) p_s(s) d\mu(\omega) \right)^2
$$

$$
= \sum_{i=1}^{K} \frac{1}{n_i} \left( \int_{\Omega} w_i(\omega) \cdot r \cdot p_r(r|s,a) \pi_e(a|s) p_s(s) d\mu(\omega) \right)^2
$$

$$
= \sum_{i=1}^{K} \frac{1}{n_i} \left( \int_{\Omega} w_i(\omega) \cdot r \, d\mu_{\pi_e}(\omega) \right)^2
$$

$$
= \sum_{i=1}^{K} \frac{1}{n_i} \langle w_i, \Pi_r \rangle_{\mu_{\pi_e}}^2.
$$

$\square$

**Theorem 5.1.** *For logging and evaluation policies $(\pi_i)_{i \in [K]}$ and $\pi_e$ that satisfy Assumption 5.1, let $\mathbf{T} \in \mathbb{R}^{K \times K}$ be the technique matrix and $\mathbf{c} \in \mathbb{R}^K$ be the contribution vector defined in (8). Let $\boldsymbol{\alpha}$ be any solution to $\mathbf{T}\boldsymbol{\alpha} = \mathbf{c}$, if one exists. Then, with respect to such $\boldsymbol{\alpha}$, the weight vector $\mathbf{w}^\star$ given by (7) solves the optimization problem (12).*

*Proof of Theorem 5.1.* We first solve the constrained minimization problem (12), initially considering only the normalization constraint: $\sum_{i=1}^{K} w_i(\omega) = 1$ for all $\omega \in \mathrm{supp}(\mu_{\pi_e})$. We will then verify that the resulting solution also satisfies the second constraint, namely $\pi_i(a|s) = 0 \implies w_i(s,a,r) = 0$ for all $i \in [K]$.

Let $\Omega := \mathcal{S} \times \mathcal{A} \times \mathcal{R}$, and define the functional $I[\mathbf{w}, \lambda]$ as

$$
I[\mathbf{w}, \lambda] := \mathrm{Var}\big[\hat{J}(\pi_e, \mathbf{w}; \mathcal{D})\big] - \sum_{i=1}^{K} \left\langle \lambda, \sum_{i=1}^{K} w_i - 1 \right\rangle_{\mu_{\pi_e}}
$$

$$
= \mathrm{Var}\big[\hat{J}(\pi_e, \mathbf{w}; \mathcal{D})\big] - \sum_{i=1}^{K} \langle \lambda, w_i \rangle_{\mu_{\pi_e}} + \langle \lambda, 1 \rangle_{\mu_{\pi_e}},
$$

where $\lambda : \Omega \to \mathbb{R}$ is the Lagrange multiplier enforcing the normalization constraint.

Next, for a perturbation direction $\boldsymbol{\delta}(\omega) := (\delta_i(\omega))_{i \in [K]}$ and $\boldsymbol{\epsilon} \in \mathbb{R}^K$, define $I(\boldsymbol{\epsilon}; \boldsymbol{\delta}, \mathbf{w}, \lambda)$ as

$$
I(\boldsymbol{\epsilon}; \boldsymbol{\delta}, \mathbf{w}, \lambda) := I[w_1 + \epsilon_1 \delta_1, \ldots, w_K + \epsilon_K \delta_K, \lambda],
$$

which can be interpreted as the variation of $I[\mathbf{w}, \lambda]$ under perturbations along the direction $\boldsymbol{\delta}$.

By the Lagrange multiplier theorem in the calculus of variations (Gelfand et al., 2000), all extrema $\mathbf{w}^\star$ of (12) satisfies the following: There exists $\lambda^\star$ such that, for any $\boldsymbol{\delta}$ and $i \in [K]$,

$$
\left. \frac{\partial I(\boldsymbol{\epsilon}; \boldsymbol{\delta}, \mathbf{w}^\star, \lambda^\star)}{\partial \epsilon_i} \right|_{\boldsymbol{\epsilon}=0} = 0,
$$

which is equivalent to

$$\left\langle \delta_i, \frac{2w_i^\star \pi_e \Pi_r^2}{n_i \pi_i} - \frac{2\Pi_r \langle w_i^\star, \Pi_r \rangle_{\mu_{\pi_e}}}{n_i} - \lambda^\star \right\rangle_{\mu_{\pi_e}} = 0. \tag{14}$$

Next, we deduce $\mathbf{w}^\star$ and $\lambda^\star$ from (14) and verify that the deduced $\mathbf{w}^\star$ is indeed a minimizer.

To proceed, let us define $\alpha_i := \langle w_i^\star, \Pi_r \rangle_{\mu_{\pi_e}}$. Then, (14) implies that, for all $\omega := (s, a, r) \in \text{supp}(\mu_{\pi_e})$,

$$\frac{2w_i^\star(\omega)\pi_e(a|s)r^2}{n_i \pi_i(a|s)} - \frac{2\alpha_i r}{n_i} - \lambda^\star(\omega) = 0.$$

Rearranging it and summing over $i \in [K]$, we obtain

$$w_i^\star(\omega) = \frac{\alpha_i \pi_i(a|s)}{\pi_e(a|s)r} + \frac{n_i \pi_i(a|s)}{2\pi_e(a|s)r^2}\lambda^\star(\omega) \tag{15}$$

$$\implies 1 = \frac{\sum_{i=1}^K \alpha_i \pi_i(a|s)}{\pi_e(a|s)r} + \frac{\sum_{i=1}^K n_i \pi_i(a|s)}{2\pi_e(a|s)r^2}\lambda^\star(\omega),$$

where we used the fact that $\sum_{i=1}^K w_i^\star(\omega) = 1$ for all $\omega \in \text{supp}(\mu_{\pi_e})$. Solving the above for $\lambda^\star(\omega)$,

$$\lambda^\star(\omega) = \frac{2\pi_e(a|s)r^2 - 2r\sum_{i=1}^K \alpha_i \pi_i(a|s)}{\sum_{i=1}^K n_i \pi_i(a|s)}.$$

Plugging $\lambda^\star(\omega)$ back into (15) results in

$$w_i^\star(\omega) = \frac{\alpha_i \pi_i(a|s)}{r\,\pi_e(a|s)} + \frac{n_i\,\pi_i(a|s)}{\sum_{j=1}^K n_j \pi_j(a|s)}\left(1 - \frac{\sum_{j=1}^K \alpha_j \pi_j(a|s)}{r\,\pi_e(a|s)}\right), \tag{16}$$

where one can easily verify that the omitted constraint "$\pi_i(a|s) = 0 \implies w_i(\omega) = 0$ for all $i \in [K]$" is satisfied for any $\boldsymbol{\alpha}$.

Next, we derive a self-consistency equation for $\boldsymbol{\alpha}$ by plugging the above expression for $w_i^\star(\omega)$ into $\alpha_i := \langle w_i^\star, \Pi_r \rangle_{\mu_{\pi_e}}$,

$$\alpha_i = \alpha_i \cdot \mathbb{E}_{\mu_{\pi_e}}\left[\frac{\pi_i(a|s)}{\pi_e(a|s)}\right] + \mathbb{E}_{\mu_{\pi_e}}\left[\frac{r\,n_i\,\pi_i(a|s)}{\sum_{k=1}^K n_k \pi_k(a|s)}\right] - \sum_{j=1}^K \alpha_j \cdot \mathbb{E}_{\mu_{\pi_e}}\left[\frac{n_i \pi_i(a|s)\pi_j(a|s)}{\pi_e(a|s)\sum_{k=1}^K n_k \pi_k(a|s)}\right]. \tag{17}$$

Since

$$\mathbb{E}_{\mu_{\pi_e}}\left[\frac{\pi_i(a|s)}{\pi_e(a|s)}\right] = \int_\Omega \frac{\pi_i(a|s)}{\pi_e(a|s)}p_r(r|s,a)\pi_e(a|s)p_s(s)d\mu(\omega)$$

$$= \int_{\text{supp}(\mu_{\pi_e})} p_r(r|s,a)\pi_i(a|s)p_s(s)d\mu(\omega)$$

$$= \mathbb{E}_{\mu_{\pi_i}}\left[\mathbb{1}[\pi_e(a|s) > 0]\right] = 1,$$

where the last equality follows from Assumption 5.1, Equation (17) can thus be compactly written as $\mathbf{T}\boldsymbol{\alpha} = \mathbf{c}$, with $\mathbf{T}$ and $\mathbf{c}$ given in (8).

Given that a solution to $\mathbf{T}\boldsymbol{\alpha} = \mathbf{c}$ exists for the matrix $\mathbf{T}$, the weights $\mathbf{w}^\star$ defined in (16), derived from any such solution $\boldsymbol{\alpha}$, satisfy the first-order condition (14). Furthermore, we show that for any solution $\boldsymbol{\alpha}$ to $\mathbf{T}\boldsymbol{\alpha} = \mathbf{c}$, which may yield different $\mathbf{w}^\star$, the resulting variance $\text{Var}[\hat{J}(\pi_e, \mathbf{w}^\star; \mathcal{D})]$ remains unchanged.

This follows from the characterization of the variance in Lemma A.2: for any $\mathbf{w}^\star$ of the form given in (16),

$$\text{Var}[\hat{J}(\pi_e, \mathbf{w}^\star; \mathcal{D})] = -\boldsymbol{\alpha}^\top \mathbf{N}^{-1}\mathbf{T}\boldsymbol{\alpha} + C,$$

where $\mathbf{N} := \mathrm{diag}(n_1, \ldots, n_K)$ and $C$ is independent of $\boldsymbol{\alpha}$. For any two solutions $\boldsymbol{\alpha}_1$ and $\boldsymbol{\alpha}_2$ to the equation $\mathbf{T}\boldsymbol{\alpha} = \mathbf{c}$, we can write $\boldsymbol{\alpha}_1 = \boldsymbol{\alpha}_2 + \mathbf{v}$ for some $\mathbf{v}$ such that $\mathbf{Tv} = 0$. Then,

$$
\begin{aligned}
\boldsymbol{\alpha}_1^\top \mathbf{N}^{-1} \mathbf{T} \boldsymbol{\alpha}_1 &= (\boldsymbol{\alpha}_2 + \mathbf{v})^\top \mathbf{N}^{-1} \mathbf{T} (\boldsymbol{\alpha}_2 + \mathbf{v}) \\
&= \boldsymbol{\alpha}_2^\top \mathbf{N}^{-1} \mathbf{T} \boldsymbol{\alpha}_2 + \boldsymbol{\alpha}_2^\top \mathbf{N}^{-1} \mathbf{Tv} + \mathbf{v}^\top \mathbf{N}^{-1} \mathbf{T} \boldsymbol{\alpha}_2 + \mathbf{v}^\top \mathbf{N}^{-1} \mathbf{Tv} \\
&= \boldsymbol{\alpha}_2^\top \mathbf{N}^{-1} \mathbf{T} \boldsymbol{\alpha}_2 + 2\boldsymbol{\alpha}_2^\top \mathbf{N}^{-1} \underbrace{\mathbf{Tv}}_{=\,0} + \mathbf{v}^\top \mathbf{N}^{-1} \underbrace{\mathbf{Tv}}_{=\,0} \\
&= \boldsymbol{\alpha}_2^\top \mathbf{N}^{-1} \mathbf{T} \boldsymbol{\alpha}_2,
\end{aligned}
$$

where the third equality follows from the symmetry of $\mathbf{N}^{-1}\mathbf{T}$ (Lemma A.1). Hence, any solution to $\mathbf{T}\boldsymbol{\alpha} = \mathbf{c}$ results in the same variance. Moreover, by Lemma A.1, the matrix $\mathbf{N}^{-1}\mathbf{T}$ is positive semi-definite. It follows that $-\boldsymbol{\alpha}^\top \mathbf{T} \boldsymbol{\alpha} \leq 0$, which implies that $\mathrm{Var}[\hat{J}(\pi_e, \mathbf{w}^\star; \mathcal{D})] \leq \mathrm{Var}[\hat{J}(\pi_e, \mathbf{w}; \mathcal{D})]$ for weights $\mathbf{w}$ corresponding to $\boldsymbol{\alpha} = 0$ (which yields the balanced IPS estimator $\hat{J}_{\mathrm{bIPS}}$ in (2)).

Therefore, the extremal weights $\mathbf{w}^\star$, derived from any solution $\boldsymbol{\alpha}$, are indeed minimizers, rather than maximizers, of the variance objective. Finally, since $\mathrm{Var}[\hat{J}(\pi_e, \mathbf{w}; \mathcal{D})]$ is a convex functional in $\mathbf{w}$ (by the characterization in Lemma 5.1), and the constraint $\sum_{i=1}^K w_i(\omega) = 1$ is convex, it follows that $\mathbf{w}^\star$ are global minimizers of the optimization problem (12). $\qquad\square$

**Lemma A.1.** *Let $\mathbf{T}$ be the technique matrix given in (8). Under Assumption 5.1, $\mathbf{N}^{-1}\mathbf{T}$ is symmetric and positive semi-definite.*

*Proof.* Recall that

$$
T_{ij} := \mathbb{E}_{\mu_{\pi_i}} \left[ \frac{n_i \pi_j(a|s)}{\sum_{k=1}^K n_k \pi_k(a|s)} \right] = \mathbb{E}_{\mu_{\pi_e}} \left[ \frac{n_i \pi_i(a|s) \pi_j(a|s)}{\pi_e(a|s) \sum_{k=1}^K n_k \pi_k(a|s)} \right],
$$

where the latter equality holds under Assumption 5.1. Hence, $\mathbf{N}^{-1}\mathbf{T}$ has matrix elements

$$
[\mathbf{N}^{-1}\mathbf{T}]_{ij} = \mathbb{E}_{\mu_{\pi_e}} \left[ \frac{\pi_i(a|s) \pi_j(a|s)}{\pi_e(a|s) \sum_{k=1}^K n_k \pi_k(a|s)} \right],
$$

which is symmetric.

Next, we show that $\mathbf{N}^{-1}\mathbf{T}$ is positive definite. For any vector $\boldsymbol{\alpha} \in \mathbb{R}^K$,

$$
\begin{aligned}
\boldsymbol{\alpha}^\top \mathbf{N}^{-1} \mathbf{T} \boldsymbol{\alpha} &= \mathbb{E}_{\mu_{\pi_e}} \left[ \frac{\sum_{i=1}^K \alpha_i \pi_i(a|s)}{\pi_e(a|s)} \cdot \frac{\sum_{j=1}^K \alpha_i \pi_j(a|s)}{\sum_{j=1}^K n_j \pi_j(a|s)} \right] \\
&= \mathbb{E}_{\mu_{\pi_e}} \left[ \frac{1}{\pi_e(a|s)} \cdot \frac{\left( \sum_{i=1}^K \alpha_j \pi_j(a|s) \right)^2}{\sum_{j=1}^K n_j \pi_j(a|s)} \right] \geq 0.
\end{aligned}
$$

$\qquad\square$

**Lemma A.2.** *For any $\mathbf{w}^\star$ of the form given in (7), let $\boldsymbol{\alpha}$ be a solution to $\mathbf{T}\boldsymbol{\alpha} = \mathbf{c}$, where $\mathbf{T}$ and $\mathbf{c}$ are defined in (8), and let $\mathbf{N} := \mathrm{diag}(n_1, \ldots, n_K)$. Then,*

$$
\mathrm{Var}[\hat{J}(\pi_e, \mathbf{w}^\star; D)] = -\boldsymbol{\alpha}^\top \mathbf{N}^{-1} \mathbf{T} \boldsymbol{\alpha} + C,
$$

*where $C$ is a constant that does not depend on $\boldsymbol{\alpha}$.*

*Proof of Lemma A.2.* Recall from Lemma 5.1 that

$$
\begin{aligned}
\mathrm{Var}[\hat{J}(\pi_e, \mathbf{w}; \mathcal{D})] &= \sum_{i=1}^K \frac{1}{n_i} \left( \left\langle w_i^2, \frac{\pi_e \Pi_r^2}{\pi_i} \right\rangle_{\mu_{\pi_e}} - \langle w_i, \Pi_r \rangle_{\mu_{\pi_e}}^2 \right), \\
&= \sum_{i=1}^K \frac{1}{n_i} \left\langle w_i^2 \Pi_r^2 \frac{\pi_e}{\pi_i}, 1 \right\rangle_{\mu_{\pi_e}} - \sum_{i=1}^K \frac{1}{n_i} \langle w_i, \Pi_r \rangle_{\mu_{\pi_e}}^2
\end{aligned} \tag{18}
$$

and recall the expression for $w_i^\star$:

$$w_i^\star(\omega) = \frac{\alpha_i \pi_i(a|s)}{r\,\pi_e(a|s)} + \frac{n_i\,\pi_i(a|s)}{\sum_{j=1}^K n_j\pi_j(a|s)}\left(1 - \frac{\sum_{j=1}^K \alpha_j\pi_j(a|s)}{r\,\pi_e(a|s)}\right),$$

where $\omega := (s, a, r)$.

We now plug the above expression for $w_i^\star$ into (18). First, we expand the term $(w_i^\star)^2\Pi_r^2\pi_e/\pi_i$ as follows:

$$(w_i^\star(\omega))^2 \cdot \Pi_r^2(\omega) \cdot \frac{\pi_e(a|s)}{\pi_i(a|s)}$$

$$= \alpha_i^2 \cdot \frac{\pi_i(a|s)}{\pi_e(a|s)} + 2r\alpha_i \cdot \frac{n_i\pi_i(a|s)}{\sum_{j=1}^K n_j\pi_j(a|s)} \cdot \left(1 - \frac{\sum_{j=1}^K \alpha_j\pi_j(a|s)}{r\pi_e(a|s)}\right)$$

$$+ \left(\frac{n_i\pi_i(a|s)}{\sum_{j=1}^K n_j\pi_j(a|s)}\right)^2 \cdot \left(1 - \frac{\sum_{j=1}^K \alpha_j\pi_j(a|s)}{r\pi_e(a|s)}\right)^2 \cdot \frac{r^2\pi_e(a|s)}{\pi_i(a|s)}.$$

By expanding and rearranging the above expression, we obtain:

$$(w_i^\star(\omega))^2 \cdot \Pi_r^2(\omega) \cdot \frac{\pi_e(a|s)}{\pi_i(a|s)}$$

$$= \underbrace{\alpha_i^2 \cdot \frac{\pi_i(a|s)}{\pi_e(a|s)}}_{\text{"Term A"}} \underbrace{-2\alpha_i \cdot \left(\frac{\sum_{j=1}^K \alpha_j\pi_j(a|s)}{\pi_e(a|s)} \cdot \frac{n_i\pi_i(a|s)}{\sum_{j=1}^K n_j\pi_j(a|s)}\right)}_{\text{"Term B"}}$$

$$+ \underbrace{\frac{\pi_e(a|s)}{\pi_i(a|s)} \cdot \left(\frac{\sum_{j=1}^K \alpha_j\pi_j(a|s)}{\pi_e(a|s)} \cdot \frac{n_i\pi_i(a|s)}{\sum_{j=1}^K n_j\pi_j(a|s)}\right)^2}_{\text{"Term C"}}$$

$$+ \underbrace{2r\alpha_i \cdot \frac{n_i\pi_i(a|s)}{\sum_{j=1}^K n_j\pi_j(a|s)}}_{\text{"Term D"}} \underbrace{- \frac{2r\pi_e(a|s)}{\pi_i(a|s)} \cdot \frac{\sum_{j=1}^K \alpha_j\pi_j(a|s)}{\pi_e(a|s)} \cdot \left(\frac{n_i\pi_i(a|s)}{\sum_{j=1}^K n_j\pi_j(a|s)}\right)^2}_{\text{"Term E"}} + C, \tag{19}$$

where $C$ is a constant that does not depend on $\boldsymbol{\alpha}$.

Next, we show that when evaluating $\sum_{i=1}^K \frac{1}{n_i}\left\langle w_i^2\Pi_r^2\frac{\pi_e}{\pi_i}, 1\right\rangle_{\mu_{\pi_e}}$, the contributions from the last two non-constant terms (Term D and E) in the above expression cancel each other. To verify this, consider the contribution from the last term (Term E):

$$-\sum_{i=1}^K \frac{1}{n_i}\left\langle \frac{2r\,\pi_e(a|s)}{\pi_i(a|s)}\frac{\sum_{j=1}^K \alpha_j\pi_j(a|s)}{\pi_e(a|s)} \cdot \left(\frac{n_i\pi_i(a|s)}{\sum_{j=1}^K n_j\pi_j(a|s)}\right)^2, 1\right\rangle_{\mu_{\pi_e}}$$

$$= -\sum_{i=1}^K\left\langle \frac{2r\sum_{j=1}^K \alpha_j\pi_j(a|s)}{\left(\sum_{j=1}^K n_j\pi_j(a|s)\right)^2} \cdot n_i\pi_i(a|s), 1\right\rangle_{\mu_{\pi_e}}$$

$$= -\left\langle \frac{2r\sum_{j=1}^K \alpha_j\pi_j(a|s)}{\left(\sum_{j=1}^K n_j\pi_j(a|s)\right)^2} \cdot \sum_{i=1}^K n_i\pi_i(a|s), 1\right\rangle_{\mu_{\pi_e}}$$

$$= -\left\langle \frac{2r\sum_{j=1}^K \alpha_j\pi_j(a|s)}{\sum_{j=1}^K n_j\pi_j(a|s)}, 1\right\rangle_{\mu_{\pi_e}}.$$

This cancels out with the contribution from the fourth term (Term D), as shown below:

$$\sum_{i=1}^{K} \frac{1}{n_i} \left\langle 2r\alpha_i \cdot \frac{n_i \pi_i(a|s)}{\sum_{j=1}^{K} n_j \pi_j(a|s)}, 1 \right\rangle_{\mu_{\pi_e}} = \left\langle \frac{2r\sum_{j=1}^{K} \alpha_j \pi_j(a|s)}{\sum_{j=1}^{K} n_j \pi_j(a|s)}, 1 \right\rangle_{\mu_{\pi_e}}.$$

Now, we compute the contributions from the second and third terms (Term B and C) in (19). The second term (Term B) contributes as follows:

$$-\sum_{i=1}^{K} \frac{1}{n_i} \left\langle 2\alpha_i \cdot \left( \frac{\sum_{j=1}^{K} \alpha_j \pi_j(a|s)}{\pi_e(a|s)} \cdot \frac{n_i \pi_i(a|s)}{\sum_{j=1}^{K} n_j \pi_j(a|s)} \right), 1 \right\rangle_{\mu_{\pi_e}}$$

$$= -2 \left\langle \frac{\sum_{j=1}^{K} \alpha_j \pi_j(a|s)}{\pi_e(a|s)} \cdot \frac{\sum_{i=1}^{K} \alpha_i \pi_i(a|s)}{\sum_{j=1}^{K} n_j \pi_j(a|s)}, 1 \right\rangle_{\mu_{\pi_e}}$$

$$= -2 \, \mathbb{E}_{\mu_{\pi_e}} \left[ \frac{\sum_{j=1}^{K} \alpha_j \pi_j(a|s)}{\pi_e(a|s)} \cdot \frac{\sum_{i=1}^{K} \alpha_i \pi_i(a|s)}{\sum_{j=1}^{K} n_j \pi_j(a|s)} \right]$$

$$= -2\boldsymbol{\alpha}^\top \mathbf{N}^{-1} \mathbf{T} \boldsymbol{\alpha},$$

Similarly, we compute the contribution from the third term (Term C) as follows:

$$\sum_{i=1}^{K} \frac{1}{n_i} \left\langle \frac{\pi_e(a|s)}{\pi_i(a|s)} \cdot \left( \frac{\sum_{j=1}^{K} \alpha_j \pi_j(a|s)}{\pi_e(a|s)} \cdot \frac{n_i \pi_i(a|s)}{\sum_{j=1}^{K} n_j \pi_j(a|s)} \right)^2, 1 \right\rangle_{\mu_{\pi_e}}$$

$$= \mathbb{E}_{\mu_{\pi_e}} \left[ \frac{1}{\pi_e(a|s)} \cdot \frac{\left( \sum_{i=1}^{K} \alpha_j \pi_j(a|s) \right)^2}{\sum_{j=1}^{K} n_j \pi_j(a|s)} \right]$$

$$= \boldsymbol{\alpha}^\top \mathbf{N}^{-1} \mathbf{T} \boldsymbol{\alpha}.$$

Finally, we verify that the contribution from the first term (Term A) cancels with the second term of (18), $-\sum_{i=1}^{K} \frac{1}{n_i} \langle w_i, \Pi_r \rangle^2_{\mu_{\pi_e}}$. The contribution from the first term (Term A) of (19) in evaluating $\sum_{i=1}^{K} \frac{1}{n_i} \left\langle w_i^2 \Pi_r^2 \frac{\pi_e}{\pi_i}, 1 \right\rangle_{\mu_{\pi_e}}$ is

$$\sum_{i=1}^{K} \frac{\alpha_i^2}{n_i} \left\langle \frac{\pi_i}{\pi_e}, 1 \right\rangle_{\mu_{\pi_e}} = \sum_{i=1}^{K} \frac{\alpha_i^2}{n_i} \int_{\mathrm{supp}(\mu_{\pi_e})} \frac{\pi_i(\omega)}{\pi_e(\omega)} d\mu_{\pi_e}(\omega)$$

$$= \sum_{i=1}^{K} \frac{\alpha_i^2}{n_i} \int_{\mathrm{supp}(\mu_{\pi_e})} \frac{\pi_i(a|s)}{\pi_e(a|s)} \cdot r \cdot p_r(r|s,a) \cdot \pi_e(a|s) \cdot p_s(s) d\mu(\omega)$$

$$= \sum_{i=1}^{K} \frac{\alpha_i^2}{n_i} \int_{\mathrm{supp}(\mu_{\pi_e})} r \cdot p_r(r|s,a) \cdot \pi_i(a|s) \cdot p_s(s) d\mu(\omega)$$

$$= \sum_{i=1}^{K} \frac{\alpha_i^2}{n_i} \int_{\mathrm{supp}(\mu_{\pi_e})} d\mu_{\pi_i}(\omega)$$

$$= \sum_{i=1}^{K} \frac{\alpha_i^2}{n_i} \int_{\mathrm{supp}(\mu_{\pi_i})} d\mu_{\pi_i}(\omega) \qquad \text{(by Assumption 5.1)}$$

$$= \sum_{i=1}^{K} \frac{\alpha_i^2}{n_i} = \sum_{i=1}^{K} \frac{1}{n_i} \langle w_i, \Pi_r \rangle^2_{\mu_{\pi_e}}, \qquad \text{(as } \alpha_i \coloneqq \langle w_i, \Pi_r \rangle_{\mu_{\pi_e}} \text{)}$$

which is the same as the second term in (18).

Therefore, we have:

$$\mathrm{Var}[\hat{J}(\pi_e, \mathbf{w}^\star; D)] = -\boldsymbol{\alpha}^\top \mathbf{N}^{-1} \mathbf{T} \boldsymbol{\alpha} + C.$$

$$\square$$

**Theorem 5.2.** *Let* $N' := \sum_{i=1}^{K} n'_i$ *with* $n'_i = mn'_i$. *Then,* $\hat{J}_{\mathrm{oIPS}}$ *is unbiased and*

$$\sqrt{N'} \left( \hat{J}_{\mathrm{oIPS}}(\pi_e; \mathcal{D}) - J(\pi_e) \right) \xrightarrow{d} \mathcal{N} \left( 0, N \operatorname{Var}[\hat{J}(\pi_e, \mathbf{w}^\star; \mathcal{D})] \right)$$

*as* $m \to \infty$, *where* $N := \sum_{i=1}^{K} n_i$.

*Proof of Theorem 5.2.* By Theorem 4.1, $\hat{J}_{\mathrm{oIPS}}$ is unbiased. To prove its asymptotic efficiency, it suffices to establish

$$\hat{J}_{\mathrm{oIPS}}(\pi_e; \mathcal{D}) := \frac{1}{N} \sum_{z \in [Z]} N^{(z)} \hat{J}^{(z)} = \frac{1}{N} \sum_{z \in [Z]} N^{(z)} \hat{J}(\pi_e, \mathbf{w}^\star; \mathcal{D}^{(z)}) + o_p(N^{-1/2}), \quad (20)$$

from which the theorem follows immediately by the central limit theorem for stratified sampling.

To this end, decompose each stratum-level estimator as

$$\hat{J}^{(z)} = \hat{J}(\pi_e, \mathbf{w}^\star; \mathcal{D}^{(z)}) + \left( \hat{J}^{(z)} - \hat{J}(\pi_e, \mathbf{w}^\star; \mathcal{D}^{(z)}) \right).$$

The second term's expectation conditioned on $\mathcal{D}^{(-z)}$ is zero as

$$\mathbb{E}\left[ \hat{J}^{(z)} - \hat{J}(\pi_e, \mathbf{w}^\star; \mathcal{D}^{(z)}) \,\Big|\, \mathcal{D}^{(-z)} \right] = \mathbb{E}\left[ \hat{J}^{(z)} \,\Big|\, \mathcal{D}^{(-z)} \right] - \mathbb{E}\left[ \hat{J}(\pi_e, \mathbf{w}^\star; \mathcal{D}^{(z)}) \,\Big|\, \mathcal{D}^{(-z)} \right]$$
$$= J(\pi_e) - J(\pi_e) = 0,$$

as both $\hat{J}^{(z)}$ are $\hat{J}(\pi_e, \mathbf{w}^\star; \mathcal{D}^{(z)})$ unbiased estimators of $J(\pi_e)$ conditioned on $\mathcal{D}^{(-z)}$.

Thus, to conclude (20), it remains to show that the conditional variance of the second term is $o_p(N^{-1})$, which implies the result by conditional Chebyshev. A direct calculation shows that:

$$\operatorname{Var}\left[ \hat{J}^{(z)} - \hat{J}(\pi_e, \mathbf{w}^\star; \mathcal{D}^{(z)}) \,\Big|\, \mathcal{D}^{(-z)} \right]$$
$$= \sum_{i=1}^{K} \frac{1}{n_i^{(z)}} \operatorname{Var}_{(s,a,r) \sim \mu_{\pi_i}} \left[ \left( \hat{w}_i^{(z)}(s,a,r) - w_i^\star(s,a,r) \right) \frac{r \pi_e(a|s)}{\pi_i(a|s)} \,\Big|\, \mathcal{D}^{(-z)} \right], \quad (21)$$

where $\hat{w}_i^{(z)}(s,a,r) := \pi_i(a|s) \rho_i^{(z)}(s,a,r)$ which is explicitly given by

$$\hat{w}_i^{(z)}(s,a,r) = \frac{\hat{\alpha}_i^{(z)} \pi_i(a|s)}{r \pi_e(a|s)} + \frac{n_i^{(z)} \pi_i(a|s)}{\sum_{k=1}^{K} n_k^{(z)} \pi_k(a|s)} \left( 1 - \frac{\sum_{k=1}^{K} \hat{\alpha}_k^{(z)} \pi_k(a|s)}{r \pi_e(a|s)} \right),$$

where $\hat{\boldsymbol{\alpha}}^{(z)} = (\hat{\alpha}_i^{(z)})_{i \in [K]}$ is the OLS estimator, given by $\hat{\boldsymbol{\alpha}}^{(z)} := (\hat{\mathbf{T}}^{(z)\top} \hat{\mathbf{T}}^{(z)})^{-1} \hat{\mathbf{T}}^{(z)\top} \hat{\mathbf{c}}^{(z)}$.

And the optimal weight $w_i^\star(s,a,r)$ (7) is

$$w_i^\star(s,a,r) = \frac{\alpha_i \pi_i(a|s)}{r \pi_e(a|s)} + \frac{n_i^{(z)} \pi_i(a|s)}{\sum_{k=1}^{K} n_k^{(z)} \pi_k(a|s)} \left( 1 - \frac{\sum_{k=1}^{K} \alpha_k \pi_k(a|s)}{r \pi_e(a|s)} \right),$$

where $\boldsymbol{\alpha} = (\alpha_i^{(z)})_{i \in [K]}$ is defined as the solution to $\mathbf{T}\boldsymbol{\alpha} = \mathbf{c}$. Since $\hat{\mathbf{T}}^{(z)}$ and $\hat{\mathbf{c}}^{(z)}$ are consistent estimators of $\mathbf{T}$ and $\mathbf{c}$, it follows that $\hat{\boldsymbol{\alpha}}^{(z)}$ is a consistent estimator of $\boldsymbol{\alpha}$, i.e.,

$$\|\hat{\boldsymbol{\alpha}}^{(z)} - \boldsymbol{\alpha}\| = o_p(1).$$

Consequently, the conditional variance in (21)) is $o_p(N^{-1})$, and thus the theorem follows. $\square$

## B    EMPIRICAL RESULTS WITH $K > 2$ LOGGING POLICIES

We extend the two-logger experiment to heterogeneous mixtures with $K \in \{3, 5\}$ logging policies. As before, we train a multinomial logistic regression model on a training split to obtain a base deterministic policy $\pi_{\text{det}}$ and set the evaluation policy to $\pi_e = \pi_{\text{det}}$. The reward is deterministic, defined as $r(s, a) := \mathbb{1}[y(s) = a]$, where $y(s)$ is the label of data $s \in \mathbb{R}^d$. The ground-truth policy value is therefore the classification accuracy of $\pi_e$ on the OPE sample.

$$J(\pi_e) = \mathbb{E}_{a \sim \pi_e(\cdot|s), p_s(s)}\big[\mathbb{1}[y(s) = a]\big].$$

We construct $K \in \{3, 5\}$ logging policies, by combining $\pi_{\text{det}}$ with the uniform policy $\pi_u$ (which ignores context and selects actions uniformly at random). For the $K = 5$ case, let $\alpha_k \in \{0.05, 0.15, 0.30, 0.70, 0.95\}$, and for $i = 1, \ldots, K$ define

$$\pi_i(a|s) := (1 - \alpha_i)\pi_{\text{det}}(a|s) + \frac{\alpha_i}{|\mathcal{A}|}.$$

The OPE sample is generated as an equal-weight mixture over these loggers, with stratum weights $w_i = 0.2$ for $K = 5$ and $w_i = \frac{1}{3}$ for $K = 3$ (so that $n_i/n = w_i$). For each example, we first draw $i \sim \text{Categorical}(w)$, then sample $a \sim \pi_i(\cdot \mid s)$, and finally assign the reward $r(s, a) := \mathbb{1}[y(s) = a]$. For the $K = 3$ case, we set $\alpha_i \in \{0.05, 0.5, 0.95\}$.

**Results.**    Across the five-logger mixtures (Figure 3 and Figure 4), our optimal IPS estimator, $\hat{J}_{\text{oIPS}}$, is consistently the most accurate. With known logging policies, it achieves Rel-RMSEs of 0.006 (SatImage), 0.004 (Letter), and 0.003 on both OptDigits and PenDigits, improving over the next-best $\hat{J}_{\text{wIPS}}/\hat{J}_{\text{bIPS}}$ and far outperforming $\hat{J}_{\text{IPS}}$. When propensities are estimated, $\hat{J}_{\text{oIPS}}$ remains the most accurate and stable, outperforming $\hat{J}_{\text{bIPS}}$, while $\hat{J}_{\text{IPS}}$ and $\hat{J}_{\text{wIPS}}$ can blow up (e.g., $\approx 30.8$ and $\approx 14.2$ Rel-RMSEs on OptDigits). This instability arises from near-zero or misestimated propensities in some strata, which induce extreme $1/\hat{\pi}_i(a|s)$ weights and heavy-tailed variance. By contrast, the sample-dependent weights of $\hat{J}_{\text{oIPS}}$ effectively control variance, yielding the most stable and accurate estimates. The same qualitative pattern holds with three logging policies (Figure 5 and Figure 6).

## C    SENSITIVITY TO THE ADDITIVE MIXTURE OF LOGGING POLICIES

We repeat the main experiments using alternative mixture ratios between the deterministic classifier policy $\pi_{\text{det}}$ and the uniform policy $\pi_u$. For $\alpha \in \{0.90, 0.75\}$, we define

$$\pi_1^{(\alpha)} = \alpha\, \pi_{\text{det}} + (1 - \alpha)\, \pi_u, \quad \pi_2^{(\alpha)} = (1 - \alpha)\, \pi_{\text{det}} + \alpha\, \pi_u.$$

All other settings—evaluation policy $\pi_e = \pi_{\text{det}}$, datasets, train/evaluation split ratio, stratum-size ratios $n_1/n_2$, number of repeats $M = 200$, and cross-fitting folds $Z = 5$—follow the main-text protocol.

**Results.**    Across both additive mixtures ($\alpha = 0.90$ and $\alpha = 0.75$), the qualitative trends of the results (Figures 3, 4, 5, and 6) remain consistent with the main-text results. Our optimal IPS estimator ($\hat{J}_{\text{oIPS}}$) achieves the lowest Relative-RMSE across almost all datasets and stratum-size ratios, in both the known- and estimated-logging-policy settings. The improvements are most pronounced in low-overlap regimes (smaller $n_1/n_2$), while the gaps narrow as overlap increases, though $\hat{J}_{\text{oIPS}}$ remains the best or tied for best throughout. Using estimated propensities increases errors uniformly but preserves the same ranking, confirming that our method is robust to mixture ratio and aligns with the tendencies observed in the main experiments.

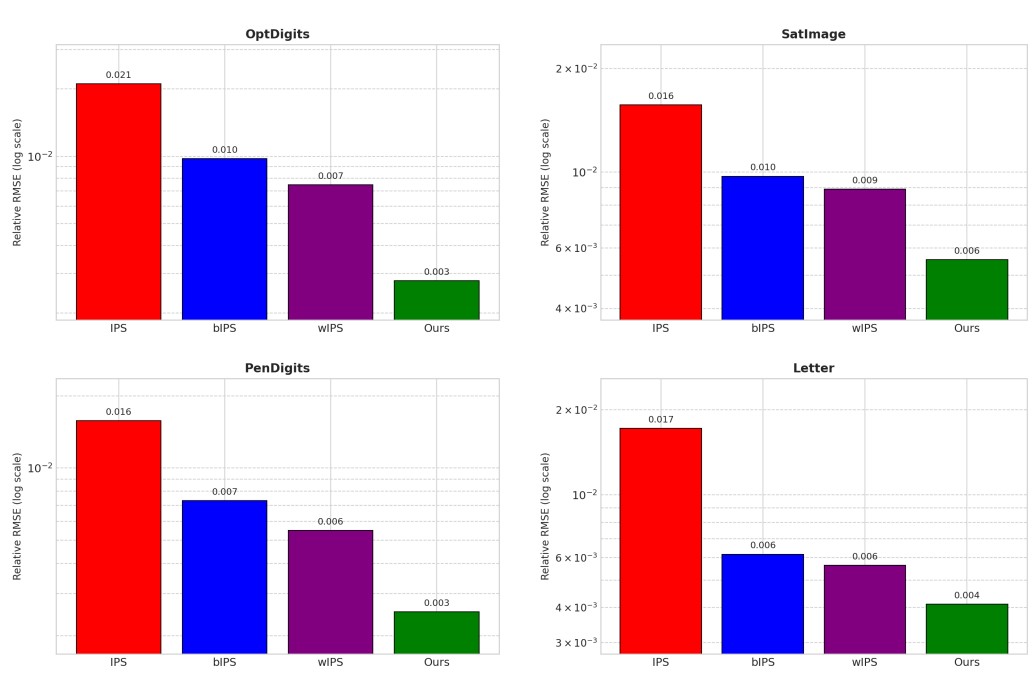

Figure 3: Rel-RMSE across five heterogeneous logging policies where they are known. Lower is better.

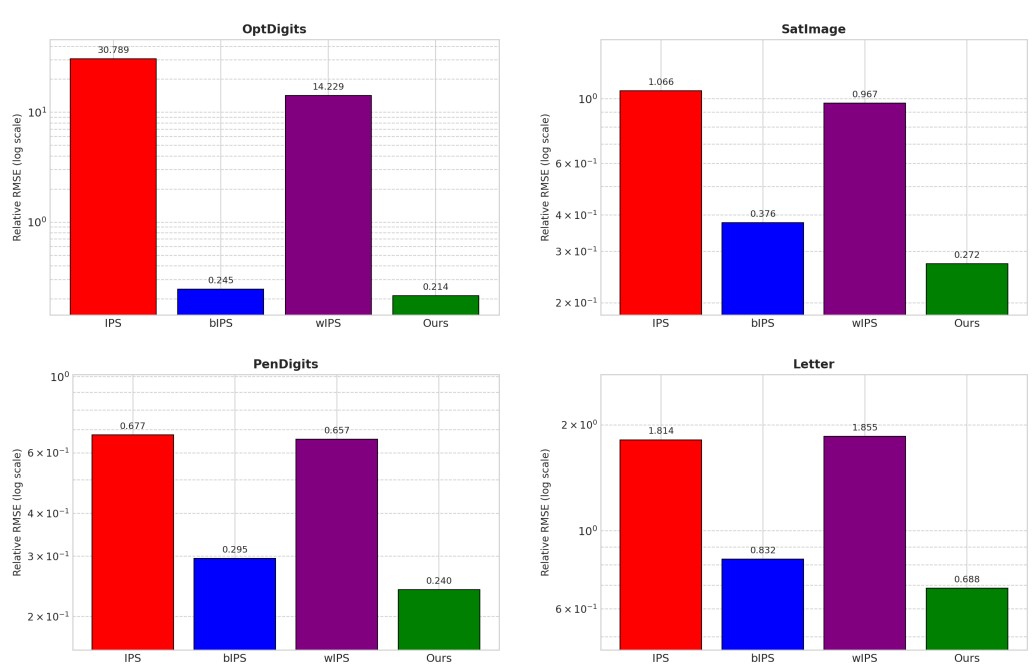

Figure 4: Rel-RMSE across five heterogeneous logging policies where they are estimated. Lower is better.

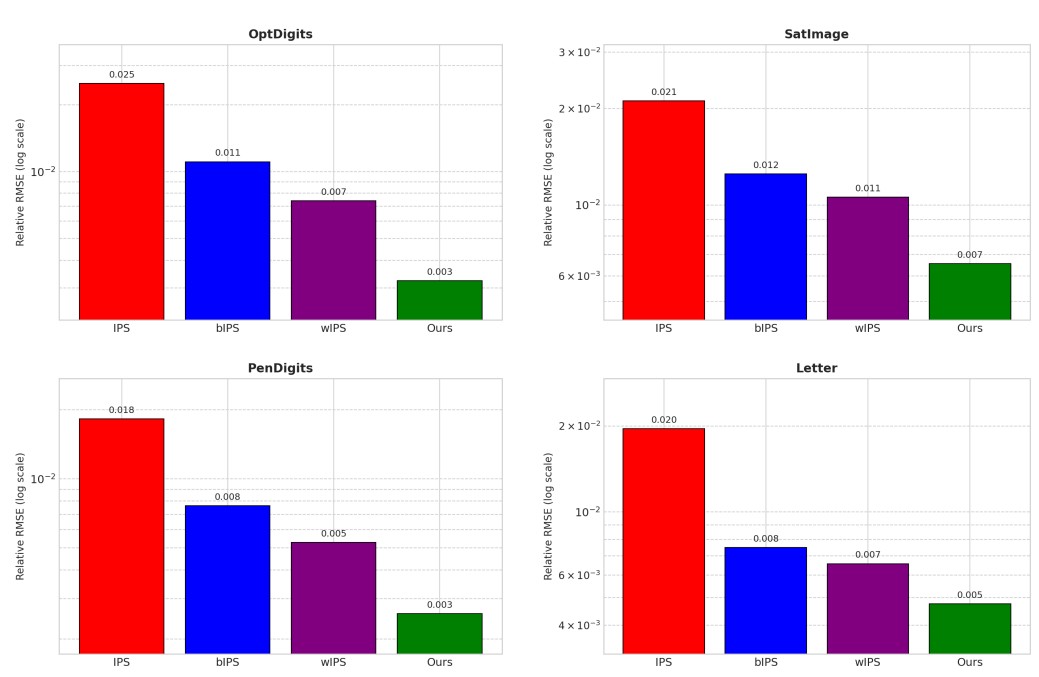

Figure 5: Rel-RMSE across three heterogeneous logging policies where they are known. Lower is better.

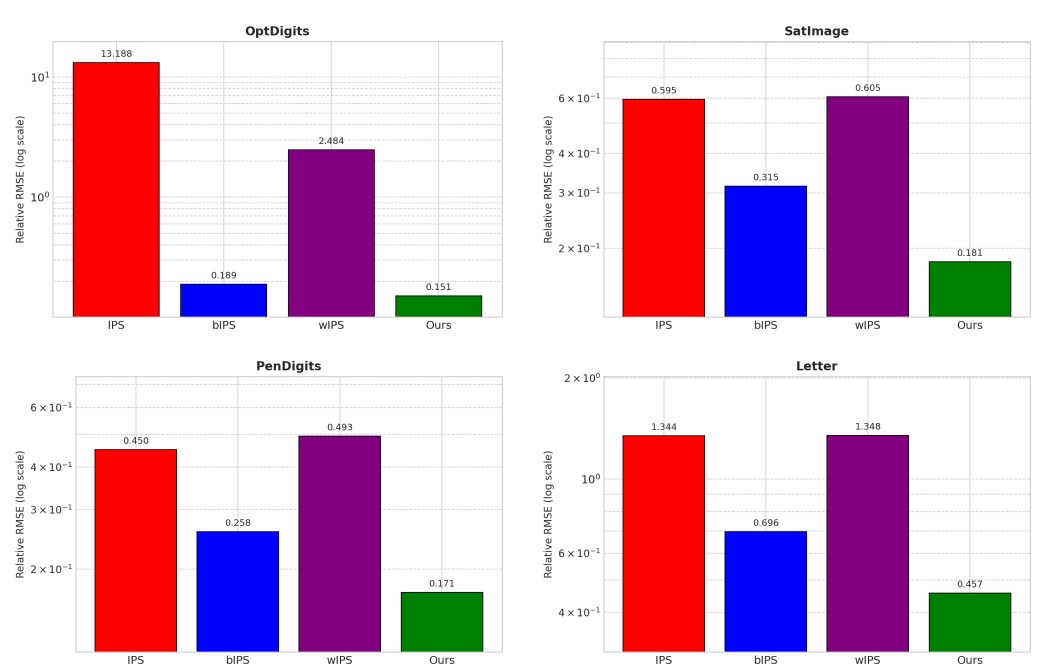

Figure 6: Rel-RMSE across three heterogeneous logging policies where they are estimated. Lower is better.

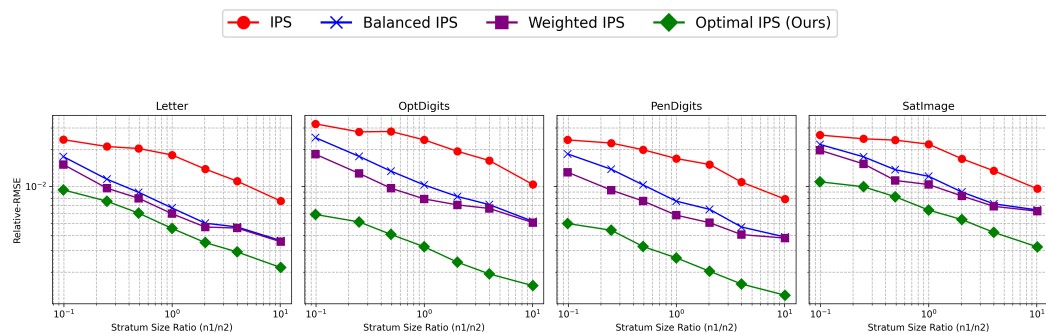

Figure 7: **Mixture** $\alpha = 0.90$. Relative-RMSEs, computed over $M = 200$ independent experiments, are reported across different datasets and estimators under the setting where the logging policies are known.

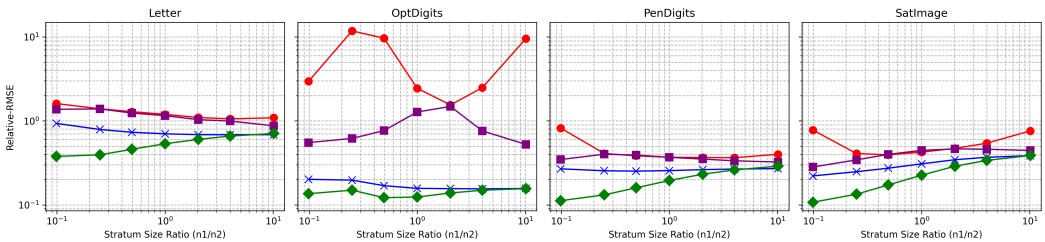

Figure 8: **Mixture** $\alpha = 0.90$. Relative-RMSEs, computed over $M = 200$ independent experiments, are reported across different datasets and estimators under the setting where the logging policies are estimated.

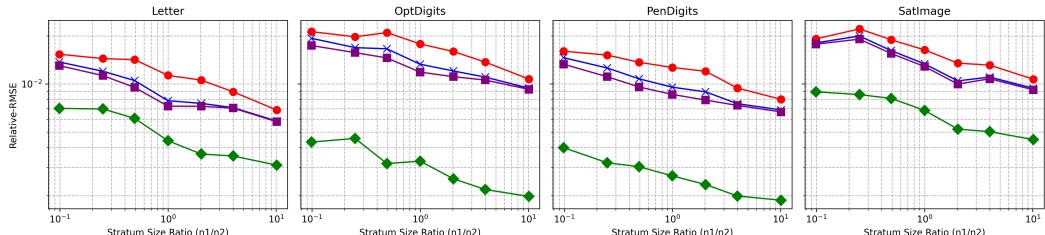

Figure 9: **Mixture** $\alpha = 0.75$. Relative-RMSEs, computed over $M = 200$ independent experiments, are reported across different datasets and estimators under the setting where the logging policies are known.

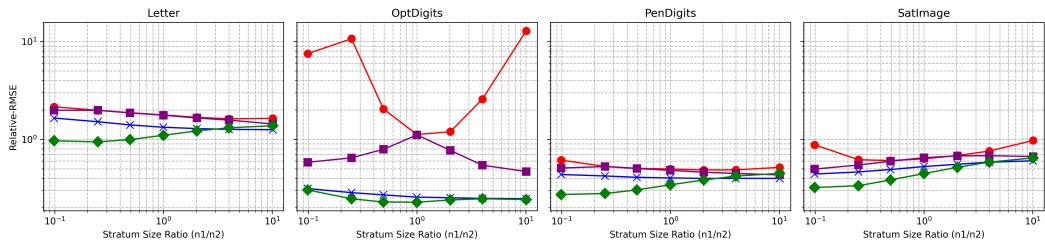

Figure 10: **Mixture** $\alpha = 0.75$. Relative-RMSEs, computed over $M = 200$ independent experiments, are reported across different datasets and estimators under the setting where the logging policies are estimated.

