# OpenReview forum: "Truly Optimal Inverse Propensity Scoring for Off-Policy Evaluation with Multiple Loggers"
_ICLR.cc/2026/Conference — ICLR 2026 Conference Withdrawn Submission_

### Official Review · Reviewer_7rWW · 2025-10-17

**Soundness:** 4
**Presentation:** 3
**Contribution:** 2
**Rating:** 6
**Confidence:** 3

**Summary:**

This paper addresses OPE in contextual bandits when logged data are collected from multiple heterogeneous logging policies. While existing IPS estimators, such as the naive, balanced, and weighted variants (Agarwal et al., 2017), each perform well under specific conditions, no single estimator has been shown to dominate in all cases.

The authors propose a novel Optimal IPS estimator (JoIPS) that derives sample-dependent weights minimizing estimator variance subject to unbiasedness. Using a calculus-of-variations approach, they characterize the optimal weights in closed form and show that they can be efficiently estimated via cross-fitting, yielding an unbiased and asymptotically optimal estimator.

Theoretical contributions include proofs of unbiasedness and asymptotic optimality. Empirically, experiments on four benchmark datasets (OptDigits, SatImage, PenDigits, Letter) demonstrate that the proposed method achieves substantial reductions in relative RMSE compared to prior IPS baselines, both when logging policies are known and when they are estimated.

**Strengths:**

The paper offers a mathematically principled derivation of optimal sample-dependent weights for the IPS estimator under multiple logging policies.

The theoretical analysis is rigorous. Theorems 4.1–5.2 establish unbiasedness, consistency, and asymptotic efficiency clearly and are backed by formal proofs in the appendix.


Experimental results are thorough, covering both known and estimated logging policies, multiple datasets, and a range of stratum ratios. The empirical trends are consistent with theory, showing robustness and variance reduction.

**Weaknesses:**

The analysis focuses on asymptotic optimality. However, in realistic OPE tasks, finite-sample variance and bias–variance tradeoffs are critical. The paper could be strengthened by including a discussion or simulation study quantifying finite-sample deviations from the asymptotic limit.

While the focus is within the IPS class, including DR or model-based estimators (e.g., Kallus et al., 2021) would help clarify whether the practical gains of JoIPS persist relative to the broader family of OPE estimators.

The mapping from classification datasets to contextual bandits uses deterministic classifiers and binary rewards. While common in OPE benchmarks, this setup limits the generality of conclusions for stochastic or real-world environments where reward noise and policy stochasticity interact.

**Questions:**

Since Kallus et al. (2021) derived efficiency bounds for DR estimators, could the calculus-of-variations framework be extended to derive “optimal doubly robust weights”? This would strengthen the connection between JoIPS and broader semiparametric efficiency theory.


Solving Tα = c via least squares is efficient for small K, but how does complexity scale for large numbers of loggers (e.g., K > 100)? Are there approximations or regularizations for high-dimensional logging mixtures?

---

### Official Review · Reviewer_ykGJ · 2025-10-25

**Soundness:** 2
**Presentation:** 3
**Contribution:** 2
**Rating:** 4
**Confidence:** 4

**Summary:**

The paper tackles off-policy evaluation with data from multiple logging policies. It introduces a generalized IPS family with sample-dependent weights (\$w_i(s,a,r)\$), derives the closed-form, variance-minimizing weights $w^\*\_i(s,a,r)$ within that family, and provides a feasible cross-fitted implementation. The resulting feasible estimator is unbiased, consistent, and asymptotically variance-optimal within the introduced generalized IPS family. On four UCI-to-bandit benchmarks, it consistently reduces the relative RMSE compared to related baselines.

**Strengths:**

**S1.** Clear formulation of the multi-logger OPE problem and of the generalized weighted IPS estimator family.

**S2.** Closed-form expression for the optimal sample-dependent weights.

**S3.** Feasible cross-fitted implementation (Alg. 1) follows best practice of keeping estimation independent of evaluation, and enjoys properties like unbiasedness, consistency, and asymptotic variance optimality within the introduced generalized weighted IPS class.

**S4.** Empirical improvements over baselines in different scenarios.

**Weaknesses:**

**W1.** Optimality here is restricted to the proposed generalized weighted IPS class. In contrast, *Kallus et al. (2021)* established global semiparametric efficiency, offering stronger theoretical guarantees, while being much more efficient computationally (see below).

**W2.** The proposed estimator is considerably more expensive: roughly speaking, I think *Kallus (2021)* scales as \$O(ZN)\$, whereas the current method scales as \$O(ZNK + ZK^3)\$, becoming costly as the number of loggers \$K\$ grows. A clock-time comparison with varying \$K\$ is needed to quantify this overhead and demonstrate scalability in practice.

**W3.** Experiments involve only two logging policies, despite the algorithm’s complexity scaling with \$K\$. Experiments do not include comparisons against *Kallus (2021)*'s estimator (they only compare to *Agarwal et al. (2017)* combined with *Kallus (2021)* feasibility trick). *Kallus (2021)* should be included as it attains the global efficiency bound. I understand that the focus here is on IPS methods and *Kallus (2021)*  is doubly robust (DR), but in practice, the goal of OPE is to minimize evaluation error (e.g., RMSE) regardless of estimator family, which is why most IPS papers include DR and direct method (DM) baselines for completeness.

**Missing references.** This is not a major weakness, but the paper omits key foundational IPS references such as [1, 2, 3] and some others. Additionally, while not directly addressing multiple loggers, regularized IPS represents a major variance reduction direction in off-policy evaluation research that warrants mention (see recent work on self-normalization, implicit exploration, exponential smoothing, pessimistic regularization, and many more [4, 5, 6, 7]).

[1] https://www.stat.cmu.edu/~brian/905-2008/papers/Horvitz-Thompson-1952-jasa.pdf

[2] https://ionides.github.io/pubs/ionides08-jcgs.pdf

[3] https://arxiv.org/pdf/1503.02834

[4] https://papers.nips.cc/paper_files/paper/2015/file/39027dfad5138c9ca0c474d71db915c3-Paper.pdf (NeurIPS)

[5] https://proceedings.neurips.cc/paper/2021/file/4476b929e30dd0c4e8bdbcc82c6ba23a-Paper.pdf (NeurIPS)

[6] https://arxiv.org/pdf/2406.03434 (UAI)

[7] https://arxiv.org/pdf/2006.10460 (AISTATS)

**Questions:**

**Q1.** The paper explicitly states only one assumption, but others may be implicit (e.g., finite variance?, the optimal weights involve $1/r$, suggesting an implicit assumption that $r \neq 0$, etc.). Since I did not read the proofs in detail, could the authors clearly list all assumptions required for the main theorems, in the same explicit manner as Assumption 5.1?

**Q2.** Section 5.2 assumes the existence of a solution to $T\alpha = c$. Could the authors discuss what conditions on the bandit environment and policies ensure the existence of such $\alpha$?

**Q3.** In line 772, the paper claims that the functional in Lemma 5.1 is convex in $w_i$. This is not immediately clear to me, could the authors eplxain why?

**Q4.** Finally, could the authors please address the requests outlined in the Weaknesses section?

---

### Official Review · Reviewer_qjNx · 2025-10-28

**Soundness:** 3
**Presentation:** 3
**Contribution:** 3
**Rating:** 8
**Confidence:** 4

**Summary:**

This paper studies how to reduce the variance of the inverse propensity scoring (IPS)-based off-policy evaluation (OPE) estimator when using data collected from multiple logging policies. Specifically, the paper first derives the conditions for enabling unbiased estimation and then derives the closed-form solution that minimizes the variance. The solution requires some empirical estimation as a plug-in component, and the paper also proposed a cross-fitting approach to estimate such plug-in parameters. The experiment results show that the proposed approach works better than other IPS-based OPE estimators for the multiple logging scenarios.

**Strengths:**

- The paper presents a nice theoretical analysis and reasoning about how to reduce the variance of OPE estimators in the multiple logging situation.

- A good point of the proposed method is that the closed-form solution is presented, and the necessary parameters can be estimated from data.

- The experiment results demonstrate that the proposed method outperforms the baselines in the accuracy of evaluation with a varying ratio of data mixing.

- Manuscript is well-written, easy to follow.

**Weaknesses:**

- While the paper provides experiments with varying ratio of data from two logging policies; an eps-greedy policy (near-deterministic) and an eps-greedy policy (near uniform random), it would be useful to see the mixture of more complex policies, e.g., softmax.

- Some qualitative analysis should be useful -- intuitively, how is the weight allocation different between the proposed method and others? Investigating what makes the difference in the variance can be insightful.

**Questions:**

- One thing I didn't understand is that the weight of each data sample depends on the indices, which are (randomly) assigned to the data. This looks a bit weird because the sample weight may change depending on the ordering of the data, while the solution should be the variance minimizer. Would this point be clarified?

- Intuitively, how can the weight allocation be different between the proposed method and others? (e.g., A toy example explaining different weight allocation can be useful.)

Note: I checked the proof of unbiasedness in the Appendix, but didn't carefully check the theoretical derivation of other parts in the paper (appeared reasonable at a glance).

---

### Official Review · Reviewer_Zykz · 2025-10-31

**Soundness:** 3
**Presentation:** 2
**Contribution:** 3
**Rating:** 4
**Confidence:** 2

**Summary:**

This paper provide a optimal ips estimator for the Off-Policy Evaluation with Multiple Loggers. The weighted IPS is built upon sample-dependent weights that minimize variance, following theoretical analysis of minimizing the variance of the IPS. They also provide a feasible solution through previous work [1]. Experimental results show the effectiveness of the proposed method.

[1] Optimal off-policy evaluation from multiple logging policies

**Strengths:**

The motivation of the paper is clear and also the method is straightforward but seems effective. The theoretical analysis and sounds and the algorithm are well-connected, where the algorithm attempts to minimize the variance of the IPS estimator. Also the experimental results are promising.

**Weaknesses:**

The presentation of the algorithm and theorem, maybe adding a few remark will help. It is a little bit hard to follow, especially on eq (7) which basically comes out of no where but without explaination which part means what.

**Questions:**

1. Regarding the setting, do you know from which logging policy the data comes? What if you do not know that?

2, just for curiosity, is it possible that this can be extended to or plug into, for example, the DR estimator or other advanced OPE methods? Does a similar property hold, or do you need to derive it from equation 12? Also, another question is that if you derive the 'optimal DR estimator' using similar technique from eq 12,  but the doubly robust estimator is already low variance estimator compared with IPS, and whether minimzing the variance  is helpfu here?

3, For the experimental results, is it possible to compare with other OPE like DR?

4, For the lemma 5.1, is there any intuitive explaination why the w can reduce the variance, especially which part of the variance will be reduced?

5, what is the performance under large shift?

---

### Note · Authors · 2025-11-23

**Comment:**

Dear Reviewers,

Thank you for the time and effort you put into reviewing our submission and for the thoughtful questions. After re-examining our core result, we identified a technical gap that cannot be fully addressed within the rebuttal period. Therefore, we have decided to withdraw the submission and work on improving the paper beyond the ICLR timeline. We sincerely appreciate your constructive feedback, which has been very helpful in shaping our research.

Best regards,
Authors of Submission 4592

**Withdrawal Confirmation:**

I have read and agree with the venue's withdrawal policy on behalf of myself and my co-authors.